# Single-molecule visualization of ATP-induced dynamics of the subunit composition of an ECF transporter complex under turnover conditions

**Solène N. Lefebvre ⬤ , Mark Nijland, Ivan Maslov & Dirk J. Slotboom ⬤ ✉**

Energy-Coupling Factor (ECF) transporters are ATP-binding cassette (ABC) transporters essential for uptake of vitamins and cofactors in prokaryotes. They have been linked to pathogen virulence and are potential targets for antimicrobials. ECF transporters have been proposed to use a unique transport mechanism where a substrate-translocating subunit (S-component) dynamically associates with and dissociates from an ATP-hydrolyzing motor (ECF module). This model is contentious, because it is based largely on experimental conditions without compartments or continuous bilayers. Here, we used single-molecule spectroscopy to investigate the conformational dynamics of the vitamin B12 transporter ECF-CbrT in membranes under vectorial transport conditions. We observed ATP hydrolysis-dependent dissociation of the S-component CbrT from, and re-association with the ECF module, in absence and presence of vitamin B12 consistent with futile ATP hydrolysis activity. The single-molecule spectroscopy experiments suggest that S-component expulsion from and re-association with the ECF module are an integral part of the translocation mechanism.

Energy-coupling factor (ECF) transporters constitute one of the seven distinct types of the ATP-binding cassette (ABC) transporters (classified as type III)[1]. They are found almost exclusively in prokaryotic organisms and are required for survival and growth as they import essential molecules such as vitamins and minerals[2]. Structurally, ECF transporters are made of four subunits, sometimes fused into multidomain proteins: a substrate-binding and translocating protein (S-component), a transmembrane scaffold protein (EcfT), and two cytosolic ATPases (EcfA and EcfA')[3]. The latter three together form the motor of the complex (termed ECF module). As opposed to other importer types in the ABC superfamily[1,4–6], the S-component of ECF transporters is not soluble but fully embedded in the membrane with the N and C-terminus located on the cytosolic side. Based on genome organization, ECF transporters have been divided into two groups: in

group I, ECF modules have a dedicated S-component, while in group II, multiple S-components can use the same ECF module to translocate their specific substrate[2,3]. In addition to phylogeny, structure and modularity, the unique nature of ECF transporters also extends to their transport mechanism, making them not only a fascinating object of study to investigate ATP-coupled processes and protein-protein interactions within the membrane, but also a promising target for novel antibiotics[7,8] as they have been linked to virulence in some pathogens[9–11].

More than a decade of biochemical, structural and molecular dynamics efforts led to several models of transport[12–14]. The models differ in the necessity of S-component dissociation from the complex as part of the transport mechanism, and in the timing of ATP binding and hydrolysis (two examples are shown in Supplementary Fig. 1).

Faculty of Science and Engineering, Groningen, Biomolecular Sciences and Biotechnology, Membrane Enzymology Group, University of Groningen, Groningen, The Netherlands. ✉e-mail: d.j.slotboom@rug.nl

Observations from cryo-EM structures[15,16], bulk activity assays[17] and molecular dynamics simulations[18] are pointing towards a thermal ratchet model (Supplementary Fig. 1a). In this model, S-components exist as solitary proteins embedded in the bilayer. Upon substrate binding, the S-component tightly associates with the ECF module. Concomitant with this association step, the S-component topples in the membrane, leading to translocation of the substrate. Subsequent ATP binding to the ATPase subunits of the ECF module results in a propagated conformational change in EcfT, deformation of the local membrane environment, and expulsion of the apo S-component[15,16,18]. ATP hydrolysis and release of ADP and $P_i$ finally reset the ECF module to the initial conformation so that an S-component can bind a new substrate. Alternative models postulate that the S-component remains associated with the ECF module, and that binding of the transported substrate is coupled to ATP binding[13], while ATP hydrolysis is used to force the transition from an outward-oriented to an inward-oriented state[12] (Supplementary Fig. 1b).

At the heart of the differences between the mechanistic models lies the question whether the subunit composition of the complex is fixed or dynamic during the transport cycle. Here, we developed a FRET sensor to study the dynamics of the complex ECF-CbrT, a vitamin B12 (cobalamin) transporter from group II, consisting of an ECF module and the vitamin B12-specific S-component named CbrT[19] (Fig. 1a). We used single-molecule spectroscopy to investigate the interaction between the S-component and the ECF module in liposomal membranes while transport across the membrane takes place (turnover conditions). By using single-molecule assays, we avoid complications in the analysis of dynamics caused by population averaging. By using liposome-reconstituted proteins, it is possible to extract mechanistic insights that are not attainable in detergent solution or lipid nanodiscs, because the detergent micelles and the membrane scaffold protein (MSP) belt affect subunit dissociation and association. We conclude that dissociation of the S-component from and re-association with the ECF module take place and that the dynamics is strictly dependent on ATP hydrolysis. Our kinetic analysis strongly suggests that the association-dissociation dynamics is part of the transport mechanism.

## Results

### Design of a FRET sensor to monitor the state of the ECF-CbrT complex

For our studies, we selected the vitamin B12-specific transporter ECF-CbrT from *Lactobacillus delbrueckii* as the expression, purification, and reconstitution in liposomes have been established in our group[19,20]. In addition, an *Escherichia coli* knock-out strain is available to screen for vitamin B12 transport in vivo[19], and a bulk fluorescence-based transport assay has been developed to measure the transport function of purified ECF-CbrT in proteoliposomes[20]. For developing a FRET sensor that reports on the complex assembly, we chose to conjugate fluorophores to engineered cysteines through thiol-maleimide reaction, for which it was necessary to create a cysteine-less background. To identify residues that could be used to replace the endogenous cysteines without loss of activity, we used the in vivo and in vitro transport assays (Supplementary Fig. 2a–g and Supplementary Table 1). We found that the transporter retained activity when three serine residues and one alanine replaced the four endogenous cysteines. Subsequently, we engineered new pairs of cysteines into the transporter for conjugation to fluorophores. In each pair of double cysteine mutants, one cysteine was located in the S-component CbrT and the other in the ECF module. In this way, we would be able to detect the potential association and dissociation of CbrT. Selection of the positions was done using information from residue distances in the resolved structure[19], accessibility of the residues, and their sequence conservation. A series of mutant transporters was generated and tested for transport activity after fluorophore labeling with Alexa Fluor 555 and

647. We chose these fluorophores because they have limited interaction with membranes, which is important for our experiments using liposome reconstituted transporters[21]. Two candidate sensors were active and were further investigated: EcfA_D77C –CbrT_L167C with the cysteine in CbrT located on the last helix (H-sensor), and EcfA_K122C –CbrT_A182C with the cysteine at the C-terminus on CbrT (C-sensor, Supplementary Fig. 3a). Both retained transport activity when reconstituted in liposomes, were specifically labeled with efficiencies of at least 68% and the fluorophore attachments were tolerated, with the H- and C-sensors showing only a slight decrease and increase, respectively, in transport activity after labeling (Supplementary Fig. 3b–d,f).

### ECF-CbrT forms a stable complex in apo condition

To establish whether the two sensors can report on the complex assembly, we first conducted single-molecule FRET (smFRET) experiments with the purified sensors solubilized in detergent solution. For this experiment, we recorded smFRET using a confocal microscope with pulsed interleaved laser excitation[22], which allows for selection of single molecules with donor and acceptor label attached, and sorting out of the molecules labeled with only donor or only acceptor fluorophores (donor-only and acceptor-only populations). This approach allows for fast screening of mutants and provides a cross-check with the Total Internal Reflection Fluorescence (TIRF) experiments used further with the sensors reconstituted in liposomes. For both sensors in detergent solution, one main FRET population was detected in the apo state (no ATP or transported substrate present) with efficiencies of 0.68 and 0.56 for the H-sensor and the C-sensor, respectively (Fig. 1b–d). In the crystal structure of detergent-solubilized ECF-CbrT[19] (pdb: 6fnp), the distance between the $C_\alpha$ atoms of the residues mutated to cysteines is 34 Å for the H-sensor which is smaller than the Forster radius $R_0$ (the inter-fluorophore distance for 0.5 FRET efficiency) of 51 Å for the pair of fluorophores used (Alexa Fluor 555 and 647). The C-terminus of CbrT is not resolved in the structure, but the distance for the C-sensor is expected to be higher than for the H-sensor. Therefore, the measured FRET efficiencies are in line with the resolved structure, and we can infer that the transporter in detergent solution adopts a full complex conformation in the apo condition. We note that conversion of FRET efficiencies into distances between the fluorophores must be treated with care because some anisotropy is measured for the fluorophores attached to the transporter (Supplementary Fig. 3e), as it has been seen on other membrane proteins[23]. Nonetheless, even with some uncertainty about the distances, the high FRET efficiencies measured show that an assembled complex is present in detergent solution, consistent with previous biochemical and structural biological characterization[19].

Both sensors were then reconstituted in liposomes composed of a mixture of *E. coli* polar lipids and phosphatidylcholine (PC) lipids. We decided to use the *E. coli* lipids even though it led to an increased fluorescence background level, because ECF transporters display a poorer activity in synthetic lipid mixtures[18] (Supplementary Fig. 2h). 1 % DOPE-biotin was added to allow for surface immobilization of the liposomes. A low protein-to-lipid ratio of 1:10.000 (w/w) was used for the reconstitution to ensure that the majority of proteoliposomes contain at most a single transporter. Liposomes were immobilized in microfluidic channels through biotin-neutravidin interaction to obtain a surface coverage where liposomes containing a labeled transporter can be spatially resolved and co-immobilization of proteoliposomes is limited (Fig. 1e and Supplementary Fig. 4a, b). Fluorescence was recorded using a home-built TIRF microscopy set-up, on which each movie was acquired for 90 s with 5 additional seconds of direct acceptor excitation at the beginning and end of the recording to allow for selection of spots containing both donor and acceptor and where photobleaching does not occur (Supplementary Fig. 5). For both sensors, a majority of extracted traces containing one donor and one acceptor dye showed static high-FRET during the 90 s with efficiency

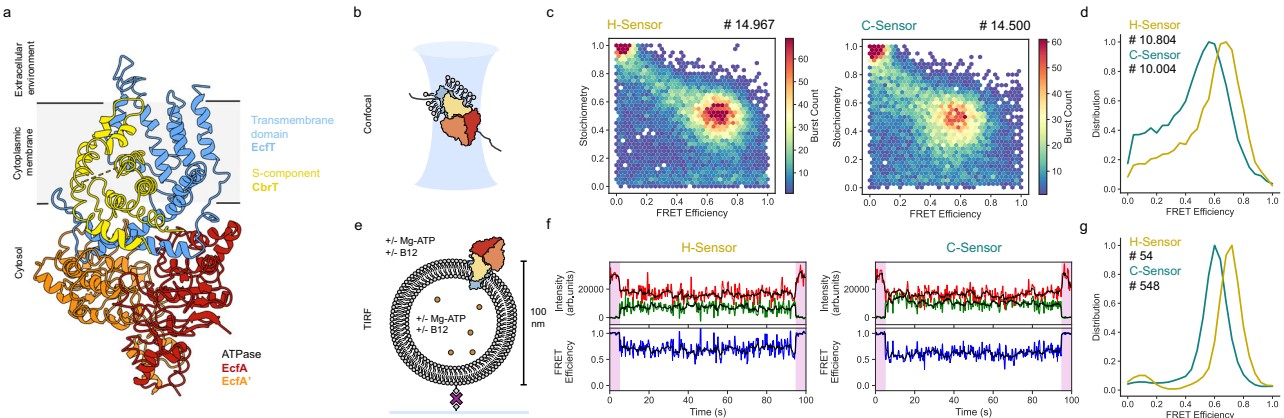

**Fig. 1 | Confocal and TIRF smFRET of ECF-CbrT sensors in apo condition.**
**a** Structure of Ecf-CbrT (pdb 6fnp) with the four subunits represented in different colors. **b** Schematic representation of confocal smFRET, an ECF-CbrT complex in detergent micelle diffuses through the confocal volume. **c** Distribution of bursts measured by confocal microscopy showing each one main FRET population for the H- and C-sensors (total number of bursts cumulated from technical triplicate recordings are indicated). **d** FRET efficiency distribution of donor and acceptor containing bursts (stoichiometry between 0.25 and 0.85) showing a maximum at 0.68 and 0.56 for the H- and C-sensors, respectively (number of bursts indicated). **e** Schematic representation of TIRF smFRET, with a proteoliposome immobilized on a PEG-coated, neutravidin-treated surface via biotinylated DOPE lipids. Biotin and neutravidin are shown as grey diamonds and a purple cross, respectively. **f** Representative TIRF traces for H and C-sensors in apo condition, top panels display the donor (green) and acceptor (red) intensities, and the bottom panels display the FRET efficiency. The colored and black lines are the signal before and after Chung-Kennedy filtering, respectively (see "Methods" section). The first and last 5 s (pink shaded areas) correspond to direct acceptor excitation. **g** Distribution of FRET efficiency for H and C-sensors (calculated from signal after CK filtering) showing a maximum at 0.72 and 0.60, respectively (total number of traces is indicated, see Supplementary Table 3 for replicates information).

values of 0.72 and 0.60 for H- and C-sensor respectively (Fig. 1f, g), consistent with in-solution measurements using confocal microscopy. It indicates that the complex in the apo state remains stable when embedded in a lipid membrane, where the protein can freely diffuse. For the rest of the study, we focused on the C-sensor, which displayed the best activity in the transport assay (Supplementary Fig. 3f).

## ATP promotes association-dissociation dynamics

To observe potential dissociation of CbrT from the complex, we used the TIRF setup with immobilized liposomes containing a single reconstituted ECF-CbrT complex. We started by testing the effect of ATP alone, since ECF transporters have been reported to display futile ATP hydrolysis in the absence of transported substrate[14,16,20]. Compared to the traces observed for the apo condition, addition of 10 mM Mg-ATP in the recording buffer led to a drastic change in the FRET efficiency distribution with the appearance of a second population with a low-FRET efficiency value close to zero (0.08) in addition to the population with higher FRET observed in apo condition (Fig. 2b). We note that in a solution of 10 mM Mg-ATP (equimolar amounts of $Mg^{2+}$ and ATP), most $Mg^{2+}$ ions are bound to ATP, and the concentration of free $Mg^{2+}$ is only ~0.7 mM[24], which is well below the concentration of free $Mg^{2+}$ that can affect liposome integrity and cause leakiness[25]. Indeed, previous bulk experiments have also shown that the use of 10 mM Mg-ATP does not lead to leakage of ATP or vitamin B12[20].

While about a fifth of the traces (19.6 ± 9.6 %) were static with the low-FRET value 0.08, a similar fraction of the traces (22.1 ± 9.1 %) were classified as dynamic (Fig. 2c, see "Methods" section). In those traces, transitions between high and low FRET states were captured, either showing a single low-to-high or high-to-low FRET transition or displaying multiple transitions within the 90 s of recording (Fig. 2a). We interpret these transitions in FRET efficiencies as CbrT dissociation from (high-to-low) and association to (low-to-high) the ECF module. The observation that only 22.1 % of traces display dynamic behavior has several potential causes. First, the reconstitution of ECF-CbrT in liposomes has been shown to result in proteins inserted in both the right-side-out and the inside-out orientation in the membrane[20], with the ATPase subunits located in the lumen of the liposome or out the

outside, respectively. Only the latter fraction was expected to respond to ATP being added on the outside. Second, we cannot exclude dynamics faster than the 200 ms time resolution in the TRIF recordings, even though no indication of fast dynamics was observed in the detergent solution. Given that the transport rates estimated from bulk experiments are low, in the order of 1 turnover per more than 100 s[19,20], it is more likely that in some cases, dynamic events occur only outside the observation time. Consistently, in about one-third of the dynamic traces, only a single transition took place during the 90 s recording. Finally, some protein complexes may have lost activity during the preparation procedure.

We repeated the experiment with Mg-ATP present on both sides of the membrane. Under this condition, not only the complexes that expose the ATPase subunits to the exterior, but also the ones oriented with the ATPases on the luminal side are accessible, and dissociation of the complexes oriented either way could take place. This condition resulted in an even larger shift to the low FRET population than with Mg-ATP added only on the outside of the liposomes (Fig. 2b), indicating that the population of transporters with the ATPase subunits located on the luminal side had also become responsive to ATP. 13.2 ± 4.4 % of traces recorded in this condition with ATP on both sides of the membrane showed dynamics (Fig. 2a, c), again either from low-to-high or high-to-low FRET, or multiple transitions. The fraction of traces showing dynamics in the observation time window was somewhat lower than in conditions with ATP added on the outside only. We do not have a conclusive explanation for this discrepancy, but speculate that it may be caused by the preparation procedure of the liposomes. To encapsulate ATP in the liposome lumen, ATP has to be present throughout the preparation steps. Therefore, dissociation of CbrT will take place already before the sizing of the liposomes (by extrusion through 100 nm filters). As a result, the dissociated S-component and ECF module may be distributed separately over the liposome population, possibly leading to two S-components or two ECF modules in the same liposome, or opposite membrane orientation of an ECF module and S-component, explaining some loss of dynamics seen when ATP is on both sides of the membrane. To avoid uncertainties related to the procedure to load liposomes with ATP, and the

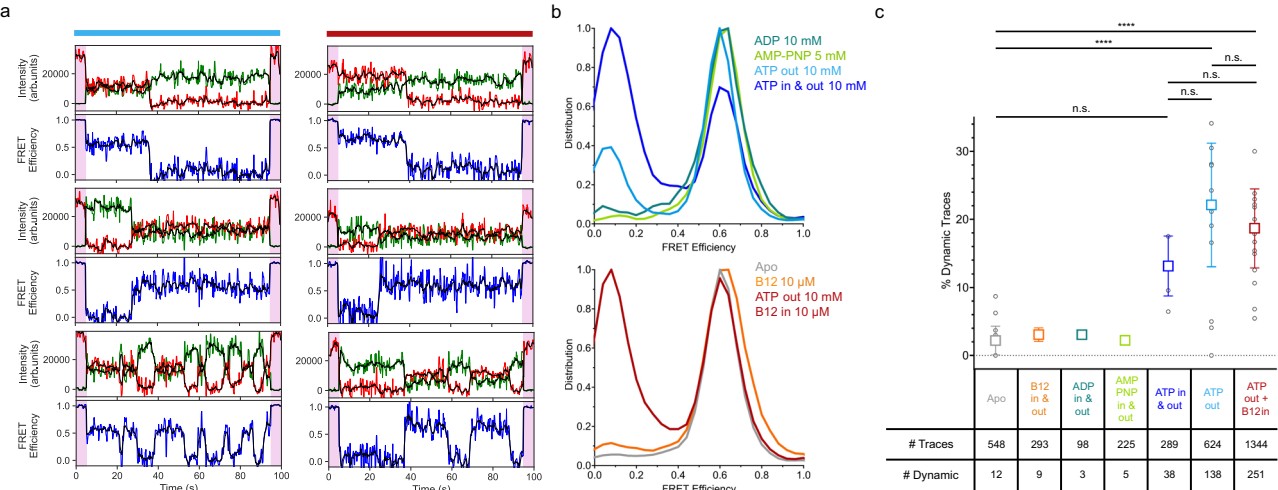

**Fig. 2 | Dynamics of the ECF-CbrT complex in liposomes using smTIRF.**
**a** Representative dynamic FRET traces showing dissociation (top panels), association (middle panels), and multiple transitions (bottom panels). The blue and red bars on top indicate the conditions without or with vitamin B12 in the lumen, respectively. In both cases, ATP was present on the outside (ATP$_{out}$). Donor and acceptor intensities and FRET efficiency are shown before (green, red, and blue, respectively) and after (black) CK filtering. The first and last 5 s (pink shaded areas) correspond to direct acceptor excitation. **b** Distribution of measured FRET efficiencies in the indicated conditions (number of traces is indicated in Fig. 2c, see Supplementary Table 3 for replicates information). **c** Percentage of dynamic traces in different conditions, the weighted average and standard deviation are shown, with the black circles representing the values for individual experiments. After a Welch's ANOVA test (right-tailed) using weighted average and standard deviation followed by a Tukey-HSD test (two-tailed), the apo condition was found to be significantly different from the ATP$_{out}$ and the ATP$_{out}$ + B12$_{in}$ conditions ($p = 0.000378$ and $p = 0.000021$ respectively), while differences between other conditions were not significant ($p = 0.063662$; $0.162573$; $0.222560$ and $0.318457$ for Apo vs. ATP$_{in\ \&\ out}$; ATP$_{in\ \&\ out}$ vs. ATP$_{out}$; ATP$_{in\ \&\ out}$ vs. ATP$_{out}$ + B12$_{in}$ and ATP$_{out}$ vs. ATP$_{out}$ + B12$_{in}$ respectively). The total number of traces for each condition is indicated in the table (See Supplementary Table 3 for replicates information).

complexity of the combined analysis of the right-side-out and inside-out oriented transporters, we decided to focus on experiments in which we add ATP only on the outside of the liposomes.

To verify that ATP hydrolysis is the driver of association and dissociation dynamics, we measured the FRET efficiency of the labeled protein complexes in the presence of Mg-AMP-PNP or Mg-ADP. Under these conditions, the FRET efficiencies were the same as in the absence of nucleotides (0.62) and remained static (Fig. 2b, c). This data shows that the transporters stayed assembled as a full complex. It is noteworthy that there was a very small percentage of traces sorted as dynamic. In the presence of ADP, three traces were classified as dynamic but were likely false positives. One trace showed an apparent dissociation, but the acceptor dye intensity at the end of the recording was half of the starting intensity, so it is likely that the spot contained multiple dyes and possibly multiple transporters, and photobleaching occurred. Two traces showed a high FRET efficiency, but elevated noise in the trace leads to a classification as dynamic because of random anti-correlation between donor and acceptor intensities. In the AMP-PNP condition, similar false positives were observed, but three traces displayed a single transition (high-to-low) interpreted as dissociation. These rare events could be caused by AMP-PNP hydrolysis, which is known to happen at a very low rate in some ATPases[26] and could allow for dissociation. Interestingly, six traces in the apo condition showed a similar pattern, indicating spontaneous dissociation, which is more difficult to explain. The local curvature and deformation of the membrane that has been observed around ECF complexes could play a role in the association and dissociation[16,18]. We speculate that extreme curvature in some liposomes could be enough to allow for dissociation of the S-component. Importantly, in both apo and AMP-PNP conditions, the observed dissociations represent only 1% of the recorded traces, and we conclude from our experiments that ATP is the main driver of the conformational dynamics in the complex causing dissociation of the S-component from the ECF module.

## S-components can be exchanged between modules
Because of the relatively high fluorescence anisotropy of the Alexa Fluor 555 and Alexa Fluor 647 fluorophores in the C-sensor (Supplementary Fig. 3e), it was essential to substantiate the interpretation that the low-FRET population represents dissociated complexes. Therefore, we performed recordings with the two single cysteine mutants that were used to create the C-sensor, labeled either with a donor fluorophore on the S-component or an acceptor dye on the ECF module. We reconstituted the labeled complexes either separately (at the same protein to lipid ratio of 1:10.000, as used for the doubly labeled complexes) or co-reconstituted the two singly labeled complexes in the same proteoliposome preparation. In the latter case, we used a protein to lipid ratio of 1:10.000 for the complex with the donor fluorophore and ratios of 1:1.000 or 1:2.000 for the complex with the acceptor fluorophore (Fig. 3b). Both types of liposome preparations were used to record smFRET in the TIRF setup. The liposome preparations containing only a single type of labeled mutant were mixed before immobilization. In this case, only co-localized liposomes on the surface would result in traces containing both donor and acceptor dye. In this sample, very few of such traces were found (8 in total), indicating that co-localization of multiple liposomes is rare. In contrast, more traces containing both donor and acceptor fluorophores were found in the co-reconstituted samples, indicating that reconstitution of two complexes in a single liposome had occurred (Supplementary Fig. 6a). In apo condition, none of the traces in the co-reconstituted samples displayed the high-FRET state recorded previously, only the low-FRET population was measured indicating that no complexes containing both fluorophores were present (Fig. 3a, c, d and Supplementary Fig. 7). Strikingly, when ATP was added to liposomes with co-reconstituted singly labeled complexes, a high-FRET population appeared and dynamic traces were observed indicating the formation of doubly labeled full-complexes (Fig. 3a, c, d). This data shows that labeled CbrT can dissociate from its unlabeled ECF-module, and re-associate with a labeled ECF-module present in the same liposomes.

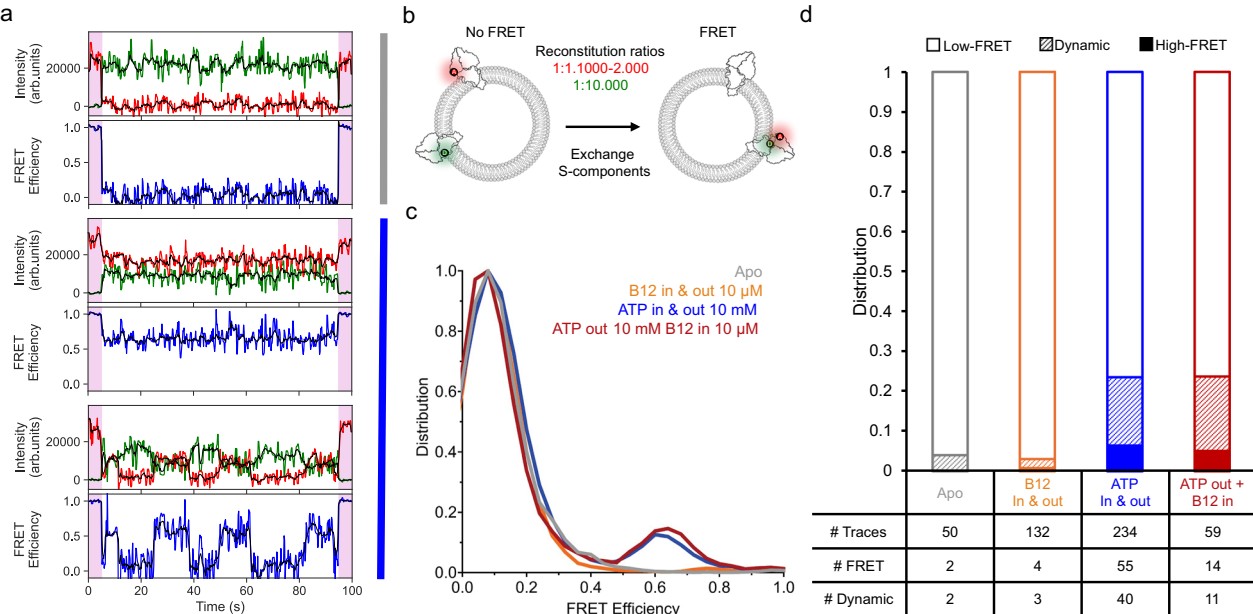

**Fig. 3 | Co-reconstitution of single-labeled ECF-CbrT complexes.**
**a** Representative traces in apo (top trace) condition or in the presence of ATP (middle and bottom traces), showing no FRET, static and dynamic FRET, respectively. Donor and acceptor intensities and FRET efficiency are shown before (green, red, and blue, respectively) and after (black) CK filtering. The first and last 5 s (pink shaded areas) correspond to direct acceptor excitation. **b** Schematic representation of the experimental conditions. Ecf-CbrT single-cysteine mutants expressed, purified, and labeled separately are co-reconstituted in the same liposomes with the indicated protein to lipid ratios (EcfA_K122C_Alexa Fluor 647 shown with the red ATPase and CbrT_A182C_Alexa Fluor 555 shown with the green S-component). **c** Distribution of FRET efficiencies in the indicated conditions. In the apo and $ATP_{out} + B12_{in}$ conditions, reconstitution with EcfA_K122C_Alexa Fluor 647 at a 1:1.000 ratio was used, the 1:2.000 was used for the ATP only and B12 only conditions. **d** Fraction of static high-FRET (solid colors), low-FRET (no color), and dynamic traces (dashed) in different conditions. Numbers of analyzed traces are shown in the table below the bars (# FRET corresponds to High-FRET + Dynamic traces), see Supplementary Table 3 for replicates information.

Consistently, none of the traces in the co-immobilized sample displayed the high FRET state regardless of whether ATP and vitamin B12 were present (turnover conditions, Supplementary Fig. 6b). This experiment confirms that S-components can swap between ECF module as bulk transport assays had suggested[17] and that the apparent low-FRET population recorded can be interpreted as dissociated complexes.

### Effect of vitamin B12 on dynamics

A notable observation from the measurements in the presence of ATP is that multiple transitions can take place during the recording of single traces (Fig. 2a), indicating that the S-component CbrT dissociates from and associates with the ECF-module even in the absence of the substrate vitamin B12. This observation is in line with previous reports on futile ATPase activity of related ECF transporters, where ATP hydrolysis takes place even in the absence of a transported substrate[14,16]. To test if futile ATP hydrolysis also takes place in ECF-CbrT, we determined the ATPase activity of purified and reconstituted ECF-CbrT (Supplementary Fig. 8a). Indeed, futile ATP turnover in these conditions was found to be over $0.5\,s^{-1}$. To investigate whether the transported substrate affects the dynamics recording by smFRET, we repeated the experiments in the presence of saturating concentrations of vitamin B12. First, smFRET TIRF recordings of the doubly labeled C-sensor complex were carried out in the presence of 10 μM vitamin B12 alone, on both sides of the liposomal membrane. Under this condition, the FRET distribution remained similar to the apo condition (Fig. 2b). From this experiment, we conclude that the vitamin B12 alone does not affect the complex assembly. Consistently, in the co-reconstitution experiment of singly-labeled ECF-CbrT complexes (Fig. 3c, d), addition of vitamin B12 to both sides of the membrane also did not change the distribution of FRET efficiencies compared to the apo condition and did not result in an increase in dynamics.

Then, we recorded the smFRET data under conditions that catalyze transport of vitamin B12 from the lumen to the outside (comparable to bulk transport assays[20], Supplementary Fig. 2f–h), with 10 μM vitamin B12 encapsulated in the lumen of proteoliposomes and 10 mM ATP in the outside buffer. Similar to the traces recorded without vitamin B12 in the lumen but in the presence of ATP on the outside ($ATP_{out}$ condition), a substantial low-FRET population is present (Fig. 2a, b). The number of traces displaying dynamic events was $18.7 \pm 5.8$ %, similar to the ATP-only condition (Fig. 2c). From this experiment, we conclude that ATP is the trigger for initiating dynamics of the complex assembly, both in the presence and absence of vitamin B12. Analysis of the traces selected out because they contain only one donor or only one acceptor show that the appearance of the low-FRET state and dynamic FRET was not a photophysical artefact from the ATP or the vitamin B12 (Supplementary Fig. 9). Consistently, also in the experiment using singly labeled, co-reconstituted complexes (Fig. 3c, d), under transport condition with ATP in the recording buffer and vitamin B12 encapsulated, the behavior was indistinguishable from that in the condition without vitamin B12 present: a high-FRET population appeared, showing that exchange of S-components took place.

Both the appearance of the low-FRET population (Fig. 2a, b), and the appearance of dynamics (Fig. 2c) upon addition of ATP, can be used as read-out to probe the response of the system to the ATP concentration. To determine apparent $EC_{50}$ values for ATP, we recorded movies at different concentrations of ATP in the recording buffer, either in the presence or absence of 10 μM of vitamin B12 in the lumen. When looking at the number of dynamic traces as a function of the ATP concentration, $EC_{50}$ values of 380 μM and 210 μM in the absence and presence of vitamin B12 were found (Fig. 4c), respectively. These values are close to the apparent $K_M$ value for ATP measured in bulk transport (190 μM, Supplementary Fig. 8b), suggesting that the dissociation of the complex is driven by ATP hydrolysis. The shift in FRET

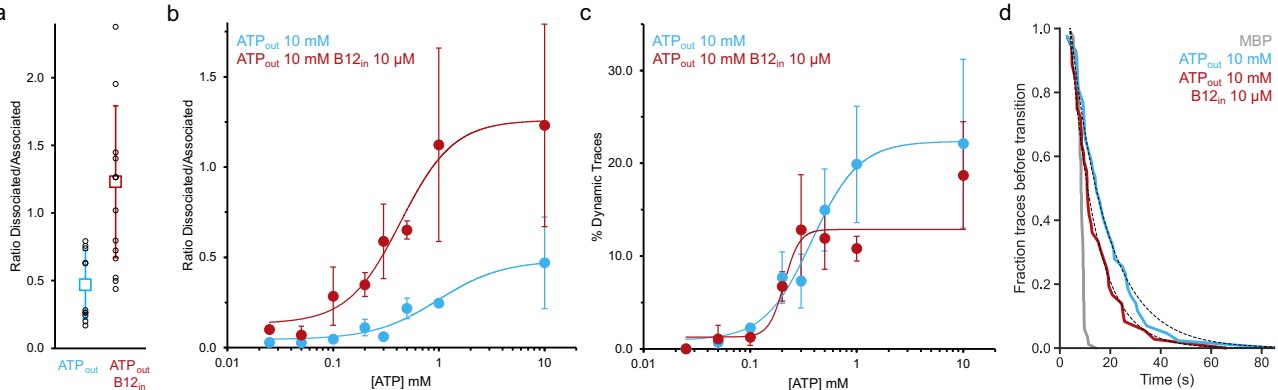

**Fig. 4 | Effect of vitamin B12 and ATP on the ECF-CbrT complex in liposomes and rate of dissociation.** Conditions in the presence or absence of 10 μM vitamin B12 in the proteoliposomes lumen are shown in red and blue, respectively. **a** Peak-to-peak ratio of the dissociated (FRET efficiency of 0.08) and associated (0.60) populations as measured in the FRET Efficiency distributions. Weighted average and standard deviation are shown, individual experiment values in black circles (See Supplementary Table 3 for replicates information). **b** ATP dependence of the dissociated/associated ratio. **c** ATP dependence of the percentage of dynamic traces. Weighted average and standard deviation are shown (See Supplementary Table 3 for replicates information). **d** Time of first dissociation event upon ATP addition. The fraction of traces for which a transition in FRET efficiency has not yet occurred after the addition of ATP in the outside buffer is shown as a function of the recording time. As a comparison, binding of maltose to maltose-binding protein (MBP) was recorded (grey). At time $t = 0$, the pump starts. The activation solution

(ATP or maltose) reaches the channel in a few seconds. Within traces classified as dynamic, dissociation was observed in 36 and 43 cases upon ATP addition to proteoliposomes with or without loaded vitamin B12, respectively (measured over 6 and 9 recording sessions, respectively). In the MBP control, a double mutant T36C/S352C labeled with Alexa Fluor 555 and 647 was immobilized, and recording was done with the addition of 200 μM maltose. A transition from 0.6 to 0.8 FRET efficiency was recorded in 86 traces corresponding to maltose binding to MBP (3 recording sessions). See Supplementary Fig. 4 for further details. A one-exponential phase decay fitting gave half-lives of 2.5 s for MBP, and 7 and 10 s for the conditions $ATP_{out} + B12_{in}$ and $ATP_{out}$, respectively. Two-tailed unpaired $t$ tests showed that the difference between decay rates exponential fits in the conditions $ATP_{out} + B12_{in}$ and $ATP_{out}$ are non-significant ($P = 0.95$), while they are both significantly slower than maltose binding to MBP ($P < 0.0001$ in both cases).

efficiency distribution in the histogram can be visualized by plotting the ratio between the associated and dissociated populations (Fig. 4a). To probe the sensitivity for ATP, this ratio was determined as a function of the ATP concentration. In this case, the $EC_{50}$ values were 980 and 430 μM (in the absence and presence of vitamin B12, respectively).

**The dynamics of association/dissociation is slower than the rate of ATP hydrolysis**

Given that the measured ATP hydrolysis turnover in ECF transporters is around 1 every 2 s (Supplementary Fig. 5b)[12,14,16,27], we wondered if every ATP hydrolysis event is strictly coupled to dissociation of the complex. To answer this question, it would be necessary to determine the dwell times in the associated and dissociated states and compare them to the ATP turnover. Unfortunately, the number of observed transitions within single movies was too low for such analysis, because of the slow associated/dissociation dynamics (Fig. 2a, c). Although an accurate quantification of the dwell times from the dynamic traces is challenging, we were able to make a tentative comparison of the dwell times with the ATP hydrolysis rate. For this, we only analyzed dwells of known duration, meaning that the first and last dwell time of each trace were not taken into account. For conditions with 10 mM Mg-ATP in the outside buffer, with and without 10 μM vitamin B12 in the lumen of liposomes, a one-phase exponential decay fitting gives dwells population with mean dwell times of 8 and 20 s for associated and dissociated states respectively (Supplementary Fig. 10), during which multiple ATP hydrolysis events are expected to take place (at a rate of 0.5 s$^{-1}$). It is important to emphasize that this dwell time analysis leads to an underestimation of the actual dwells, because dwells that last beyond the recording window are not used, and traces without transitions or with only a single transition are not used. At the same time, the bulk ATPase activity measurements are averaged over the entire population, assuming 100 % reconstitution efficiency, so the effective hydrolysis rate is likely faster. Overall, the dwell time analysis strongly supports an interpretation that not every hydrolysis event is efficient in

dissociating the S-component. Apparently, for every dissociation event of the S-component, several unsuccessful attempts take place.

While the addition of B12 did not affect the percentage of traces showing dynamics, it did lead to a difference in the distribution of FRET efficiencies in the presence of ATP, with a larger dissociated population when vitamin B12 is present (Figs. 2b, d, and 4a). To probe the underlying cause of this shift, it would be necessary to determine accurate dwell times of the associated and dissociated states, which was not possible (see above). However, it was possible to obtain an approximate comparison of the dwell times in the associated state in the presence and absence of vitamin B12. For this, we collected movies in which we started the recording in the absence of ATP, where the complexes are associated, and then added ATP at a defined timepoint to capture the time it takes for the first dissociation event to happen (Supplementary Fig. 3a). We compared the timing of the first dissociation event with the timing of maltose binding to immobilized maltose binding protein (MBP), which had been labeled with fluorophores to act as a FRET sensor for maltose binding[28]. These experiments show that binding of maltose on MBP occurs approximately 10 s after the start of the pump, and is highly synchronized throughout the population (Fig. 4d and supplementary Fig. 3c–e), which is expected for a high-affinity binding reaction with fast $k_{on}$ ($2.3 \times 10^7$ M$^{-1}$.s$^{-1}$)[29] and $k_{off}$ of 90 s$^{-1}$ (which at equilibrium leads to a homogeneous signal in the 200 ms time windows used for data averaging). The first dissociation of ECF-CbrT also happens around 10 s, but there is much more variability in the population of immobilized molecules in the timing of the event, with transitions happening throughout the entire 90 s of movie recording. The kinetics of dissociation of complexes in the presence and absence of vitamin B12 were both significantly different from the maltose binding kinetics, but not significantly different from each other (see legend of Fig. 4d for quantification of the exponential decays and statistics), suggesting that the difference in the distribution of FRET efficiencies is not caused by differences in the dwell times of the associated state.

## Discussion

We developed a sensor that allows for the direct visualization of single-molecule dynamics of the ECF transporter for vitamin B12, ECF-CbrT, reconstituted in proteoliposomes, under turnover conditions. The sensor contains a pair of fluorophores (Alexa Fluor 555 and Alexa Fluor 647), of which one is attached to the ECF module, and the other to the S-component CbrT. Importantly, the labeled transporter complex retained wild-type ATP-dependent transport activity for vitamin B12 when purified and reconstituted in liposomes. In the fully assembled transporter (containing both the ECF module and CbrT), the two fluorophores are at a distance that gave rise to high-FRET efficiency (0.62), while in the dissociated state, the ECF module and the S-component were able to diffuse away from each other in the continuous bilayer of the liposome, effectively eliminating FRET. When purified and reconstituted in the apo state (in the absence of nucleotides and vitamin B12), the complex remained fully assembled. The complex also remained stable upon the addition of Mg-ADP, Mg-AMP-PNP, or vitamin B12. In contrast, in the presence of Mg-ATP, dissociation of CbrT from the ECF module was triggered, and dynamics (repeated association-dissociation cycles) started to occur. The data is consistent with models in which dissociation is an integral part of the translocation mechanism (Supplementary Fig. 1a), and highlights the relation between ATP hydrolysis and the dissociation of the S-component from the ECF-module. It is noteworthy that the association-dissociation dynamics takes place both in the presence of vitamin B12 (when transport takes place), and in its absence. Dynamics in the absence of vitamin B12 is indicative of futile ATP hydrolysis cycles, which has been reported for other ECF transporters in the past. We quantified the futile ATP hydrolysis rate of ECF-CbrT to be ~ $0.5\,s^{-1}$, consistent with previous measurements in other ECF transporters[12,14,16]. Futile hydrolysis of ATP has also been observed in some other ECF transporters[13,14,30] but is not a conserved feature of all ABC transporters, where in many cases the transported substrate has been shown to stimulate ATPase activity[31].

Although accurate determination of the dwell times in the associated and dissociated states was not possible because of the relatively small numbers of transitions within each recording (consistent with the low transport rates determined in bulk assays), we could measure transporter populations in the associated state with dwell times longer than expected for the ATP hydrolysis rate (mean dwell times of 7.9 and 8.7 s, Supplementary Fig. 10). The full complex thus remains stable for seconds which indicates that not every hydrolysis is efficient to expel the S-component. We speculate that it is possible that ATP hydrolysis triggers disturbances in and deformation of the membrane that only occasionally allow for dissociation of CbrT. Membrane deformations have been observed in MD simulations and structural studies, and the importance of the membrane deformation in the association/dissociation process has been hypothesized[16,18]. Other (artificial) causes of membrane disturbance, for instance, high membrane curvature in very small liposomes, might also lead to infrequent S-component dissociation, possibly explaining the rare dissociation events seen without ATP.

While the results show that dissociation of the S-component takes place in an ATP-dependent way, it does not formally show that expulsion of the S-component is essential for transport to occur. To probe this, it would be necessary to combine the measurements on association and dissociation dynamics (presented here) with measurements of transport at the single-molecule level[32], by using a sensor for vitamin B12[20,33]. Such simultaneous detection of conformational dynamics and transport requires at least three-color imaging, and will be explored in the future. However, although not formally conclusive, the kinetics of dissociation and association observed here, and the turnover numbers for transport of vitamin B12 determined from ensemble experiments (~ 0.01 per second) are consistent with the notion that transport and dissociation-association dynamics are linked.

ECF-CbrT is an example of a group II ECF transporter, in which other S-components than CbrT can also use the same ECF module to translocate their respective substrates[2]. Under physiological conditions, the ECF module is expressed at low, constitutive levels, and the S-components may be upregulated to high levels, depending on the environmental availability of each cognate substrate[34]. For group II ECF transporters, it has been shown that S-components for different substrates compete for association with the limiting amount of ECF modules[17,34,35]. Competition in some cases is more effective when the substrate is bound. For instance, pioneering work from Henderson and colleagues[34] in whole cell assays using *Lactobacillus casei* showed that the thiamin- and folate-bound S-components ThiT and FolT, respectively, inhibit biotin uptake by the ECF transporter more effectively than the apo-ThiT and apo-FolT. Based on these experiments, it was concluded that the dissociation of S-components from the limited number of ECF modules present in a cell is required to allow for competitive transport of different substrates, and that substrate-binding to an S-component facilitates subsequent association with an available ECF module. This interpretation has recently been substantiated using purified proteins co-reconstituted into liposomes[17]. However, substrate binding to other S-components does affect their competitive ability for association with the ECF module. For instance, in the same study by Henderson, biotin had no effect on folate and thiamin uptake. The data presented here on the dissociation of CbrT from and reassociation with the ECF module are consistent with the competition model. The data also reveal that the presence of vitamin B12 under turnover conditions favors the dissociated state, with the complexes residing on average more time in the dissociated state than when vitamin B12 is absent (Fig. 4a). Because the dissociation of CbrT from the ECF module is not affected by vitamin B12, it is likely that the association is disfavored in the vitamin B12-bound state, suggesting that vitamin B12 binding to CbrT will decrease the competitive ability for association with the ECF module, a prediction that can be tested in the future.

While dissociation of the S-component from the ECF module is strictly dependent on ATP (Fig. 2), the dynamics of dissociation and re-association occur regardless of the presence of the transported substrate vitamin B12. These continuous cycles of association and dissociation seem energetically costly when no substrate is translocated. In addition the occurrence of multiple ATP hydrolysis events to achieve a successful dissociation of the S-component, the possibility of association with an apo S-components, and, in the group II of ECF transporters, the competition between different S-components with the same module, all likely contribute to the observed futile ATP hydrolysis in ECF transporters. Although energetically costly, the frequent expulsion of S-components may increase the chances of binding a substrate-loaded S-component, as well as varying the type of substrate transported. Therefore, it would be interesting to compare the association and dissociation dynamics in an ECF transporter from group I or in group II, the competition between CbrT and other S-components to evaluate their respective affinities for the ECF-module, and investigate, at the single-molecule level, the change in dynamics when different S-components are in competition.

We speculate that a benefit may be associated with the futile cycle of the ECF transporter for vitamin B12, where it may be needed to allow for transport of the scarce nutrient that binds with very high affinity to the transporter. The apparently inefficient use of ATP may be inherent to transporters that bind the transported substrate with very high affinity[36,37]. In this respect, it is noteworthy that for the uptake of vitamin B12 by humans, the substrate is captured with high affinity by the protein Intrinsic Factor, which is subsequently internalized by receptor mediated endocytosis, and transferred to lysosomes[38]. To

release the substrate from Intrinsic Factor, the protein is proteolytically degraded in the lysosome, which comes at the expense of hundreds of equivalents of ATP needed for protein synthesis[39]. Apparently, the vitamin B12 substrate is so valuable that systems associated with such high cost have evolved. Nonetheless, for ECF-CbrT, there may be other mechanisms of regulation in the cells (at the expression level[34] or other proteins interacting with the complex[40]) to reduce the energetic expense.

We conclude that smFRET experiments are suitable to study these dynamic steps in transport mechanisms, but challenging to apply in a lipid bilayer environment, with only a handful of reported cases[33,41,42]. The data presented here indicate that dissociation of the CbrT from, and association with the ECF module is most likely part of the transport mechanism for vitamin B12. Dissociation strictly depends on ATP hydrolysis, and futile ATP hydrolysis is inherent to the transport mechanism. For a comprehensive quantitative analysis of the slow kinetics of the dissociation-association cycle, it will be necessary to circumvent the limitation from the fluorophores photostability in order to visualize single transporters during longer time periods.

## Methods

### Cloning
For Expression, a p2bad vector was used, containing downstream its first arabinose inducible promoter the ECF operon from *L. delbrueckii* made of EcfA (uniport: Q1GBJ0); EcfA' (uniport: Q1GBI9) and EcfT (uniport: Q1GBI8) in frame with a N-terminal deca-histidine tag and a TEV protease site (ENLYPQG). CbrT or FolT2 genes from *L. delbrueckii* (Uniport: Q1G7W0 and Q1G929, respectively) were inserted downstream of the second arabinose inducible promoter. Site-directed mutagenesis for cysteine substitutions was performed by PCR with complementary or 3'-overhang primers on the ECF operon inserted in a simple pbad vector and CbrT in a pACYCara vector. Mutated genes were sub-cloned into p2bad using XbaI and XhoI restriction enzymes for CbrT and EcorI and BspEI for the ECF operon. Mutation insertion and correct sub-cloning were verified with Sanger sequencing by Eurofins Genomics. The resulting plasmids were transformed into chemically competent *E. coli* MC1061 cells for expression and into *E. coli* ΔFEC strain cells for growth assays. Bacterial strains, constructs, and primers used in this study are listed in Supplementary Tables 1 and 2.

### Growth assay
The vitamin B12-dependent growth assay was performed with the vitamin B12-deficient *E. coli* ΔFEC strain as previously described[19,20,43]. Pre-cultures of the strains carrying expression vectors of ECF-CbrT mutants were grown in M9 medium supplemented with 0.00001% (w/v) L-arabinose and 50 µg.mL⁻¹ methionine at 37 °C for 24 h. Cells were diluted into 200 µL M9 medium supplemented with 0.00001% (w/v) arabinose in the presence of either 50 µg.mL⁻¹ methionine or 1 nM vitamin B12 (cyano-cobalamin) and grown in 96-well plates (Greiner) at 37 °C. The plates were sealed with sterile and gas-permeable foil, and the optical density at 600 nm ($OD_{600}$) was measured every 10 min on a microplate reader (Gen5, BioTek) for 36 h. The measurements were performed as biological triplicates, containing each technical triplicates. Lag times λ were calculated in GraphPad Prism (version 10.0.2) from the averaged growth curves with the Gompertz-fit[44].

### ECF-CbrT expression and membrane vesicles preparation
ECF-CbrT mutants in p2bad transformed into *E. coli* MC1061 cells were pre-cultured overnight in LB before inoculation at 1/100 in 2 L of LB broth supplemented with 100 µg.mL⁻¹ ampicillin in 5 L baffled flasks. Cultures were grown at 37 °C, 190 rpm, and induced at $OD_{600}$ around 0.8 by 0.01 % arabinose (w/v). After 3 h of expression cells were harvested by centrifugation (15 min, 6.000 × g, 4 °C) and resuspended in resuspension buffer (50 mM potassium phosphate (KPi) pH 7.5, 20 %

glycerol (w/v)). Cells were diluted in breaking buffer (50 mM KPi pH 7.5, 20% glycerol, 1 mM MgSO4, 0.2 mM PMSF (Roth), and DNase I (Sigma-Aldrich)) to an $OD_{600}$ below 200 and lysed by two passages through a high-pressure homogenizer HPL6 system (Maximator) at 25 kpsi. Cells debris was removed by centrifugation (15 min, 27.000 g, 4 °C), and membrane were collected by ultra-centrifugation (120 min, 185.500 × g, 4 °C). Membrane pellet were homogenized in resuspension buffer using a potter, flash frozen, and stored in aliquots at −70 °C.

### ECF-CbrT purification
Membranes were solubilized at a total protein concentration of 5 mg.mL⁻¹ (determined using a bicinchoninic acid assay kit, Thermo Fisher Scientific) for 1 h at 4 °C on a nutating platform in solubilization buffer (50 mM KPi pH 7.5, glycerol 10 % (w/v), 300 mM NaCl, 1 % (w/v) dodecyl-β-maltoside (DDM, Glycon), 1 mM DTT and 15 mM Imidazole). DTT was omitted from all buffers when purifying cysteine-less mutants. Un-solubilized material was removed by ultracentrifugation (30 min, 444.000 × g, 4 °C) and supernatant was incubated for 1 h nutating at 4 °C with nickel resin (Ni Sepharose 6 fast flow, Cytiva) pre-equilibrated with 10 column volumes (CV) of equilibration buffer (50 mM KPi pH 7.5, glycerol 10 % (w/v), 300 mM NaCl, 0.05 % (w/v) DDM, 1 mM DTT and 15 mM Imidazole). Packing of the resin in a polypropylene column (PolyPrep, Bio-Rad) was done while allowing unbound protein to flow through. Further removal of un-specifically bound protein was done using 20 CV of washing buffer (50 mM KPi pH 7.5, glycerol 10 % (w/v), 300 mM NaCl, 0.05 % (w/v) DDM, 1 mM DTT and 50 mM Imidazole) and purified protein was eluted in 3 fractions (0.9, 1.5 and 1 CV) in Elution1 buffer (50 mM KPi pH 7.5, glycerol 10 % (w/v), 300 mM NaCl, 0.05 % (w/v) DDM, 1 mM DTT and 300 mM Imidazole). The second fraction was further purified using size-exclusion chromatography (SEC, Chromelab, Biorad) with a Superdex 200 increase 10/300 GL column (Cytiva) equilibrated with SEC buffer (50 mM KPi pH 7.5, 150 mM NaCl, 0.05% (w/v) DDM, 1 mM DTT). The peak fractions corresponding to the purified protein were pooled and used either for labeling or directly for liposome reconstitution.

### ECF-CbrT fluorophore labeling
Purified protein was loaded on a nickel resin equilibrated in SEC buffer (typically 1.5 nmol of protein for 0.2 mL of resin in a 1.2 mL Bio-spin column, Bio-Rad) and flow-through was re-applied twice. DTT was washed away by 10 CV of labeling buffer (50 mM KPi pH 7.5, 150 mM NaCl, 0.05% (w/v) DDM). 50 nmol aliquots of Alexa Fluor 555 and Alexa Fluor 647 (Thermo Fischer) were dissolved in 0.02 CV of DMSO, added to 2 CV of labeling buffer, and loaded on the resin as quickly as possible after DTT removal. Labeling was done for 2.5 − 3 h at 4 °C in the dark without nutation. Un-reacted dye was removed in the flow-through and by washing with 10 CV of labeling buffer. Elution2 buffer (50 mM KPi pH 7.5, 300 mM NaCl, 0.05 % (w/v) DDM, and 300 mM Imidazole) was applied, and the eluting labeled protein was identified visually and collected. Labeled protein was used for proteoliposome reconstitution, and the rest was passed through a desalting column (NAP5, Cytiva) equilibrated with labeling buffer to get rid of imidazole. Some of the collected fractions were aliquoted, flash frozen after addition of glycerol (final concentration 10 % (v/v)) and store for in solution recording, the rest was used to record an absorption spectrum in a UV-VIS spectrophotometer (Cary 100 Bio) to calculate the efficiency of labeling. Labeling efficiencies in different labeling batches were between 68 and 98 %. Labeling of purified MBP was done in a similar way, excluding DDM from buffers. Anisotropy was measured for free dyes in water, in buffer (KPi 50 mM pH7.5, 150 mM NaCl, 0.05% (w/v) DDM), and in 50% glycerol. For the same dyes within C-sensor solubilized in the protein buffer, anisotropy was measured without ligands (apo) or with 10 mM Mg-ATP. Measurements were performed on a Jasco FP-8300 scanning spectrofluorometer (Spectra Manager, Jasco Inc., Easton, Maryland) with polarized filters previously described[23].

## Reconstitution in liposomes

Lipids were prepared as previously described[45]. *E. coli* polar lipids were prepared from *E. coli* total extract (Avanti lipids) by acetone pre-cipitation and extraction with diethyl ether. Egg phosphatidylcholine lipids and DOPE-biotin lipids (Avanti lipids) in chloroform were mixed to the dried *E. coli* polar lipids at a weight ratio of 3:1:0.04 (*E. coli* polar:PC:DOPE-biotin). After chloroform evaporation in a rotavapor, the lipid film was dissolved at 20 mg.mL$^{-1}$ in 50 mM KPi pH 7 buffer, lipids were aliquoted and stored in liquid nitrogen. Prior to protein reconstitution, lipids were extruded through a 400 nm polycarbonate filter (Avestin), diluted to 4 mg.mL$^{-1}$ in 50 mM KPi pH 7.5 buffer, and destabilized by the addition of Triton X-100 until half of the initial absorbance at 540 nm was measured. ECF-CbrT was added to the destabilized lipids at protein to lipid ratios of 1:400 for transport assays and 1:10.000 for smFRET unless stated otherwise. After 30 min incubation at 4 °C, nutating, the detergent was gradually removed by 4 steps of Bio-Beads SM-2 (Bio-rad) addition. Proteoliposomes were collected by ultracentrifugation (30 min, 444.000 × *g*, 4 °C), resus-pended in 50 mM KPi pH 7.5 buffer at a lipid concentration of 40 mg.mL$^{-1}$, and stored aliquoted in liquid nitrogen.

## Preparation of the transport sensor BtuF488

The sensor was prepared as previously described[20]. BtuF D141C mutant from *E. coli* was expressed and purified similarly as ECF-CbrT with the following modifications. Expression was done at 25 °C, cell lysis was done in the presence of 1 mM DTT, and the supernatant obtained after removal of the cell debris was directly incubated with 2 mL bed volume of nickel resin pre-equilibrated with 20 CV of equilibration buffer (50 mM KPi pH7.5, 300 mM NaCl, 10% (v/v) glycerol and 1 mM DTT). Packing of the resin into an Econo-Pac column (BioRad) was done while allowing unbound proteins to flow through. Further removal of un-specifically bound protein was done using 40 CV of wash buffer (50 mM KPi pH 7.5, 300 mM NaCl, 50 mM imidazole, 10% (v/v) glycerol, 1 mM DTT). BtuF was eluted from the column with elution buffer (50 mM KPi pH 7.5, 300 mM NaCl, 350 mM imidazole, 10% (v/v) gly-cerol, 1 mM DTT). Fractions containing most protein were further purified by SEC in SEC buffer (50 mM KPi pH 7.5, 200 mM NaCl, 10% (v/v) glycerol, 1 mM DTT). 21 nmol BtuF was rebound to 0.3 mL bed volume of nickel resin, that was pre-equilibrated with 20 CV SEC buf-fer. DTT and glycerol were removed with 9 CV labeling buffer (50 mM KPi pH 7.5, 200 mM NaCl), after which the protein was immediately incubated at 4 °C for 2 h with a 2-fold molar excess of AF488 C5 mal-eimide (Jena Science) or XDF-488 C5 maleimide (AAT Bioquest) dis-solved in labeling buffer. Un-reacted fluorophore was removed with 20 CV washing buffer, followed by the elution of the labeled protein (50 mM KPi 7.5, 200 mM NaCl, 350 mM imidazole, 5% (v/v) glycerol). Labeled protein was further purified by SEC in SEC buffer (50 mM KPi pH 7.5, 150 mM NaCl, 5 % (v/v) glycerol). Aliquots were flash frozen in liquid nitrogen and stored at − 70 °C until further use.

## Bulk fluorescence transport assay

Bulk fluorescence transport assays with BtuF$_{488}$ was previously described in details[20]. In this study, proteoliposomes in assay buffer (50 mM KPi, pH 7.5, and 150 mM NaCl) were loaded with 20 μM of vitamin B12 (cyano-cobalamin) through five freeze-thaw cycles, fol-lowed by extrusion through a 400 nm polycarbonate filter (Cytiva). Excess of vitamin B12 was removed by two steps of dilution in assay buffer and ultracentrifugation (30 min, 444.000 × *g*, at 4 °C). Trans-port reactions were performed in black polystyrene 96-well plate (Greiner) using proteoliposomes resuspended in assay buffer at a lipid concentration of 2.5 mg.ml$^{-1}$ in a final volume of 200 μl. The sensor BtuF$_{488}$ was supplemented to the external buffer with a concentration of 60 nM (unless stated otherwise). Mg-ATP or Mg-ADP (Sigma Aldrich) was added to a final concentration of 10 mM to trigger transport (or a lower concentration for the $K_M$ determination

experiment). The transport activity was immediately monitored on a microplate reader (Spark 10 M, TECAN) by measuring the fluorescence signal every 20 s at an excitation wavelength of 485 nm (bandwidth 5) and an emission wavelength of 520 nm (bandwidth 10) using the instruments' software (SparksControl version 2.3, TECAN). The mea-surements were performed at 30 °C, and signals were normalized to the first reading after the addition of nucleotides.

## Coupled enzyme ATPase activity assay

The ATPase activity of ECF-CbrT (cysless) was determined in liposomes (lipid to protein ratio of 1:250) with a size of 400 nm using an enzyme-coupled spectrophotometric assay[14,46]. The assay was performed in a 96-well plate (Greiner) with a final volume of 200 μL containing 50 mM KPi, pH 7.5, 150 mM NaCl, 0.3 mM NADH (Roth), 4 mM phosphoe-nolpyruvate (Roth), 3.5 μL pyruvate kinase-lactic dehydrogenase solu-tion (Sigma-Aldrich) corresponding to 2.1 to 3.5 units pyruvate kinase and 3.2 to 4.9 units lactic dehydrogenase and 10 mM Mg-ATP and 3.2 μg of protein. The decrease of NADH was monitored over time by measuring the absorbance at 340 nm on a microplate reader (Spec-traMax ABS plus, SoftMax Pro Version 7.1.2, Molecular Devices) at 23 °C. The concentration of NADH was subsequently calculated by applying the Beer-Lambert law using the extinction coefficient of 6.22 cm$^{-1}$ μM$^{-1}$ at 340 nm. The pathlength of light through the solution was determined by measuring the absorbance (A) of 200 μL water at 900 and 977 nm and using the formula ($A_{977}$ − $A_{900}$)/0.18. The mea-surements were performed as biological duplicate containing each technical triplicates. The data was corrected for 50% protein orientation.

## In-solution single-molecule confocal recording

In-solution smFRET recording were done with Pulsed Interleaved Excitation (PIE) at 40 MHz and 20 MHz on a MicroTime 200 confocal microscope (PicoQuant). Prior to the recording, microscope slides (170 μm thickness, No. 1.5H precision cover slides, VWR) were coated for at least 1 min with 1 mg.mL$^{-1}$ bovine serum albumin (BSA) after which the BSA solution was replaced by 150–200 μL of sample at around 300 pM in recording buffer (KPi 50 mM pH7.5, 150 mM NaCl, 0.05 % (w/v) DDM). Fluorophores were alternately excited using a 532 nm (LDH-P-FA-530-B) and a 638 nm (LDH-D-C-640) laser. The laser beam was focused 7 μm above the glass-solution interface, with an oil-immersed objective lens (UPlanSApo 100 × 1.40 NA, Olympus). The emitted photons from the sample were coordinated through a 100 μm pinhole, separated through a laser beam splitter (ZT640RDC; Chroma Technology), filtered by emission filters (either HQ690/70, Chroma Technology or 582/75, Semrock) and recorded by two single-photon counting modules (donor photons: SPCM-AQRH-14-TR, acceptor photons: SPCM-CD-3516-H; Excelitas Technologies). 10–30 min acqui-sitions were done using the SymPhoTime 64 software (Picoquant).

## Flow-cell and proteoliposomes immobilization

Microfluidic channels were made using soft lithography. Poly-dimethylsiloxane (PDMS, Sylgard 184, Dow) was poured on a wafer (4-channel pattern in SU-8 photoresist on silicon master, Supplementary Fig. 3), and after polymerization, inlet and outlet were created using punches (Harris uni-core sampling tools 1 and 1.2 mm, respectively). PDMS and microscopy glass slide (170 μm thickness, No. 1.5H precision cover slides, VWR) were sonicated in isopropanol, then assembled together after plasma treatment (Plasma Etch Unitronics), and poly-ethylene tubing was added in the inlet and outlet (PE20 and PE60, respectively, BD Intramedic). After 30 min of 1 M KOH treatment, channels were coated with PLL-PEG at 1 mg.mL$^{-1}$ containing 5 μg.mL$^{-1}$ of PEG-PLL-Biotin (SuSoS) for 45 min. Proteoliposomes at a lipid con-centration of 8 mg.mL$^{-1}$ were prepared by 3 freeze-thawing cycles and extrusion through a 100 nm polycarbonate filter in buffer (50 mM KPi pH7.5, 150 mM NaCl plus potential substrates or nucleotides to

encapsulate). Before immobilizing the proteoliposomes, channels were incubated with 0.5 mg.mL$^{-1}$ BSA and 0.01 mg.mL$^{-1}$ Neutravidin (Thermofischer) using a syringe pump (New Era NE-1000X). Proteoliposomes were diluted 4-fold and incubated in the channel for about 30 s before wash with 0.5 mL buffer to remove non-immobilized liposomes and if applicable, to remove encapsulated substrate from the external buffer.

### Single-molecule TIRF recording

smFRET was recorded using a house-built, objective-based, TIRF setup with a high numerical aperture objective (UAPON 100x TIRF, NA 1.49 oil, Olympus) on an inverted microscope (IX71 Olympus). Emission light was separated using a dual-channel simultaneous-imaging system (DV2 photometrics), and donor and acceptor fluorescence was detected by an EMCCD digital camera (C9100-13, Hamamatsu). Movies of 100 s were recorded at a 200 ms frame rate using ImageJ and the Micro-Manager user interface (version 2.0). Excitation was done using a 532 nm laser (OBIS LS 532 nm, Coherent), and acceptor presence was detected by 5 s of direct excitation at the beginning and end of the movie using a 637 nm laser (OBIS LX 637 nm, Coherent). Filtered and degassed recording solutions were pumped at 50 µL.min$^{-1}$ during imaging and were composed of 50 mM KPi pH7.5, 150 mM NaCl, an anti-blinking agent (1 mM Trolox, Merck), and an oxygen scavenging system made of 50 nM Protocatechuate 3,4-Dioxygenase (Merck) and 2 mM Protocatechuic acid (Merck).

### smFRET confocal data analysis

smFRET confocal data was analyzed using the Python package FRETbursts (version0.7.1)[47] and python scripts (Python version 3.6.13, NumPy version 1.1.19, pandas version 1.1.5, Matplotlib version 3.3.4, lmfit version 1.0.3) adapted from published codes[23] after file conversion using phconvert (version 0.9.1)[48]. Background rate was calculated with exponential fitting every 30 s (calc_bg), and a burst search was done with the default parameters (burst_search(m = 10, F = 6)). Correction factors for crosstalk between donor and acceptor detectors and direct acceptor excitation[49] were calculated on donor-only and acceptor-only populations, and bursts containing a minimum of 40 photons (total photon count including acceptor photons in acceptor excitation periods) were plotted in a 2D FRET Efficiency-Stoichiometry distribution plot[47]. To exclude donor only and acceptor only molecules, bursts with a stoichiometry between 0.25 and 0.85 were selected, and the normalized FRET efficiency distribution was plotted with a 0.04 bin size.

### smFRET TIRF data analysis

smFRET TIRF traces were extracted using a custom ImageJ plugin after movies were corrected for donor/acceptor channels alignment, camera electronic offset, and laser profile[50]. Spots were selected from the donor channel using an intensity threshold, and the background area was defined in both channels. Traces were further processed using custom Python codes (Python version 3.12, NumPy version 1.26.4, SciPy 1.13.1, Matplotlib version 3.8.4, Seaborn version 0.13.2). First, a three or seven-frames median filter was applied to the signal and background values, respectively. Then, the signal was corrected using background subtraction, correction for crosstalk between donor and acceptor channels and for direct acceptor excitation. Traces likely containing only one donor and one acceptor dye without photobleaching were selected using intensity constraints in the first and last 10 s of recording (threshold values and correction factors were determined using samples containing donor only or only acceptor molecules). Selected traces were further de-noised using Chung-Kennedy filtering[51] with parameters KC = M = 10 and p = 1. To sort static FRET traces and traces containing dynamic events, we calculated the correlation between donor and acceptor intensities, normalized by the squared average total intensity. Using a threshold of − 0.01 on the resulting value allowed to classify correctly the majority of traces as dynamic or static (high and low FRET) while minimizing false positives and false negatives (tested on a subset of data, > 90 % of traces annotation matched the classification when transitions were detected by eye). FRET efficiency values after Chung-Kennedy filtering of selected traces were combined by recording conditions and plotted in a frequency distribution plot with a 0.04 bin size (GraphPad Prism). Static high-FRET traces of the assembled complex were detected when the median FRET efficiency value over the donor excitation period was above 0.4. To determine the dwell times, the FRET efficiencies after CK filtering of dynamic traces were loaded on the MASH FRET[52] software, and transitions were detected using the STaSi method, and the transition plot and state configuration to extract dwell times was done using a binning of 0.01 and the GM method. Statistical tests were performed using GraphPad Prism (t-tests, one-way ANOVA, Tukey-HSD) or customed Python scripts (Welch 's ANOVA, Tukey-HSD; Python version 3.12, NumPy version 1.26.4, SciPy version 1.13.1, pandas version 2.2.2). Figure were prepared using GraphPad Prism, ChimeraX [https://www.rbvi.ucsf.edu/chimerax](version 1.4)[53] and python scripts (Python version 3.12, NumPy version 1.26.4, Matplotlib version 3.8.4, Seaborn version 0.13.2). Example of smFRET and confocal data are available through the open repository Zenodo (https://doi.org/10.5281/zenodo.15118562).

### Reporting summary

Further information on research design is available in the Nature Portfolio Reporting Summary linked to this article.

## Data availability

The previously solved structure of Ecf-CbrT is available through the Protein Data Bank (PDB) under the accession code 6FNP. Protein sequences used in this study are available through UniProt under the accession codes P37028 for BtuF, Q1GBJ0 for EcfA, Q1GBI9 for EcfA', Q1GBI8 for EcfT, Q1G292 for FolT2, and Q1G7W0 for CbrT. Examples of confocal recordings and TIRF traces are available through the open repository Zenodo (https://doi.org/10.5281/zenodo.15118562). Source data are provided in this paper.

## Code availability

Scripts used to process confocal and TIRF smFRET data are available on Github at https://github.com/MembraneEnzymology/smFRET and through the open repository Zenodo (https://doi.org/10.5281/zenodo.15118562).

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

## Acknowledgements

We want to thank Joris Goudsmits for the initial building of the TIRF set-up and inputs in recording and analysis of the TIRF data. We also thank Marco van den Noort for providing us with the purified cysteine mutant of MBP and for inputs on the smFRET confocal recording and analysis. D.J.S. acknowledges funding from NWO (TOP Grant 714.018.003). S.N.L. acknowledges support from the Federation of European Biochemical Societies (FEBS) through funding of a long-term fellowship. I.M. acknowledges funding from the European Union under the HORIZON TMA MSCA Postdoctoral Fellowships action (project MemProDx, 101149735).

## Author contributions

D.J.S. and S.N.L. designed the research. S.N.L. performed cloning, growth assays, single-molecule experiments, and data processing and analysis. S.N.L. and M.N. prepared the transport sensor $BtuF_{488}$ and performed the transport assays. M.N. preformed the couple enzyme activity assay. M.N. prepared the sample, and I.M. performed the anisotropy measurement. S.N.L. and I.M. optimized the smFRET TIRF data analysis. S.N.L. and D.J.S. prepared the manuscript with inputs from all authors.

## Competing interests

The authors declare no competing interests.
