## [Transparent Peer Review file · Nature Communications]

Single-molecule visualization of ATP-induced dynamics of the subunit composition of an ECF transporter complex under turnover conditions

Corresponding Author: Professor Dirk Slotboom

Version 0:

Reviewer comments:

Reviewer #1

(Remarks to the Author)

Lefebvre et al present an informative and well designed work offering the direct observation of ATP-induced dynamics of vitamin B12 transporter ECF-CbrT under turnover conditions. The work primarily relies on single molecule FRET, confocal and camera based, as well as readout of activity for the transport in detergent and reconstituted in liposomes. Based on these readout authors find that the dynamics are dependent explicitly on ATP hydrolysis and are observed in the presence but also absence of vitamin B12. They also find that S-component expulsion and reassociation are an integral part of the translocation mechanism. The experimental work is nicely designed, the quality is sound, multiple control experiments are performed and the data support the authors claim in general. The work is stellar for the field, nicely written and in general easy to follow and important for a wide audience .

Addressing the technical elements below will help authors to validate statistically the claims.

1. Authors conclude that ATP, and not vitamin B12, alters dissociation and association dynamics of the complex. Their conclusion is primarily relied on reporting minimal shift in FRET distribution and no change in the number of dynamic traces in the presence/absence of B12 . There are a few important elements to be clarified for this central message to be carried out by the experiments :

a) The graph in 3d shows very similar % of dynamic/(non dynamic) in all 4 conditions of apo, B12, ATP and ATP+B12 ($11/14=0.78\%$, and $40/55=0.73\%$). This may be impaired by the very low statistics. Authors should comment on this
b) The figure 4d displays the decay with time of the % traces before transition. Authors claim that the B12+ATP trace is similar to the ATP only. However, the two decay appear not identical as authors write. To support this claim authors can provide a statistical test (either decay rates with exponential decays, or other appropriate models) and output a “t” test or KS test of their significance

2. In the last paragraph of results authors write “that not every hydrolysis of ATP is coupled to release of the S-component” and they justify this by comparing the turnover rate of 2-3 sec⁻¹ to the dwell time of transitions in fig2ac. The manuscript would greatly benefit if this is not qualitatively but quantitative. Authors in this case have enough traces to extract dwell times using HMM analysis of the SM FRET traces . they could use any of the existing tools/software that are for example in the recent work lead by S Schmidt and the FRET community in Nature commun 23

3. Fig 1 confocal smFRET , why is the background in the 2D graphs so high ? it appears as noise is integrated., what are the thresholds utilized in the confocal data ?

4. Authors should comment on how do they exclude potential variation in chromophore intensity readouts by the bilayer interaction and by the potential orientation of the ECF-CbrT complex . Do authors expect any effect of the orientation of the protein relative to liposomes to the actual dynamic reported here ?

5. Labeling efficiency is ~68% . this would result in 3 out of 10 protein to be unlabeled. While this is easy to correct for intermolecular recording , it can affect the readout of the intermolecular FRET, at least the rates.

6. Authors report FRET efficiencies of 0.68 and 0.56 for the H-sensor and the C-sensor respectively, and compare with Ca distances of 35 Å. They could use the Monte Carlo simulation toolbox reported by Seidel group in Nat Methods 12, to accurately calculate from FRET efficiencies to Ca distances

7. Authors write they used 1:10,000 (w/w) ratio to ensure 1 protein per liposome. A typical liposome of e.g. 50 nm contains 50,000 lipids and if the above ratio is indeed the one used and if all protein is reconstituted, it would result to higher likelihood of containing a protein per liposome (actually more like 5). Authors should check and comment on that

8. The two FRET distributions recorded by confocal and TIRF have very different widths. Authors should comment and explain this

9. None of the figures describes if data are from biological or technical replicates. Authors should provide statistical test to validate their conclusion for example in Figure 2E but also in other places. Also the number of technical and biological replicates is not clearly stated in some figures.

Minor comment

10. The experiments with the maltose should be adequately described as if someone is not familiar with the type will not understand them,

L139, sentence is unclear

L62 deformation of membrane, are there references?

The article would benefit for an additional schematic representation (or a zoom with a colour code) where all the components are shown.

L215, There seems like there is a double "the"

The figure legend of figure 4 is unclear (especially for b and c).

Reviewer #2

(Remarks to the Author)

Summary

I thank the authors for their work and effort, for pursuing the challenges of the study and for the time invested in compiling the manuscript. I appreciated this work, as it represents a rare extension of single-molecule FRET investigations to ABC transporters of less conventional architecture like ECF transporters and their unique membrane-embedded substrate binding (S-) components. The authors attempt to dissect mechanistic key aspects of the turnover cycle and its ligand dependencies with strategies based on various schemes of intra- and inter-molecular FRET. I found particularly elegant the rationale of using half-labeled transporter complexes to directly reveal that the S-component can interact dynamically and switch between various ECF modules.

Furthermore, this work attempts to elucidate ambiguous postulates of proposed models of transport mechanism for this specific transporter type. I think this acquires even more value considering that, as the authors highlight, one of the many therapeutic strategies proposed to fight pathogenic bacterial species and develop antimicrobials is based on targeting ECF transporters, due to their involvement in microbial virulence.

Major Comments

However, after carefully reading the manuscript, I identified a few points and concerns that I would like the authors to address and comment on before I can recommend publication. These points are listed below:

1. The authors emphasize that the one major obstacle towards the postulation of a unanimous transport model for ECF transporters is the lack of studies where vectorial translocation of cobalamin can be mimicked. Thus, the authors present for the first time single-molecule observations of type III ABC transporters (ECF) conformational dynamics conducted in compartments separated by a lipid bilayer, else proteoliposomes. The authors stress on the importance of having established a vectorial system for this specific transporter type. I would like the authors to elaborate on the advantages introduced by this novelty in the approach for studying the ECF transporters. This, particularly because my understanding is that the findings of this manuscript are in accordance to related methodologies previously employed by others to study the transport system of interest.

2. In the second paragraph of the results section, the ATP role in association and dissociation dynamics of S-component from ECF module is characterized. The authors observe about 22% of traces exhibiting FRET transitions at the presence of 10 mM ATP and 10 mM Mg²⁺. They identify two causes to rationalize the observed low percentage of dynamic traces. While I find plausible the speculation about the low turnover rate impacting the occurrence of dynamic traces, I don't think the authors are working under conditions where ligands ATP and Magnesium ions are only selectively accessible to the sub-population of inside-out oriented transporters. Pillar biochemical studies conducted on ABC (and other) transporters (Liu et al., J Biol Chem, 1997; doi: 10.1074/jbc.272.35.21883) report that magnesium induces a lipid phase transition (Allen et al., Biochemistry, 1990; doi: 10.1021/bi00464a013) which causes liposomes permeabilization at concentration of 10 mM or higher. Furthermore, the working principle of the well-established NADH-coupled assay (<https://microbiology.ucdavis.edu/heyer/wordpress/wp-content/uploads/2013/11/ATPase-assay.pdf>) to assess

spectrophotometrically the total ATP hydrolytic activity of transporters relies on the use of 10 mM Magnesium with the purpose to permeabilize lipid vesicles. I don't think the orientation of the transporter is one plausible cause to explain the low percentage of observed dynamic traces. I would like to understand the rationale of the authors on this matter.

3. On a related note and to my knowledge, magnesium can introduce photophysical phenomena such as dye blinking at 10 mM (or higher) concentration. Therefore, it could represent a source of errors in the estimate of FRET efficiency and distance distribution. It would be ensuring if the authors could provide evidence, by which they are able to show that magnesium does not introduce artifacts in the observations (e.g., false FRET transitions). For example, this could be achieved by performing control experiments with dsDNA standard labeled with donor and acceptor pairs located at known distances, immobilized on surface and monitored at the presence and at the absence of magnesium ions. Alternatively, the apo state of ECF-CbrT at the presence and absence of magnesium (without ATP) could be recorded.

4. Page 6, line 182: The authors comment on the lower percentage of dynamic traces observed when ATP and Mg²⁺ are entrapped in the lumen of the liposomes. Their interpretation for this unexpected finding is that S-components spatially segregate from ECF modules, leading to vesicles containing multiple ECF or S-components. If that is true, it should be relatively easy to corroborate this interpretation with the quantification of traces containing more than one single molecule under these two conditions (vesicles with ATPout and ATPout/in).

The authors also debate that the futile cycles in the case of the "ATPout/in" condition due to ATP depletion can explain the observed lower dynamics. I don't think this is a valid argument, because ultimately ATP is provided externally under similar conditions as in "ATPout". Therefore, the percentage of dynamic traces should be at least equivalent to the dynamics found for ATPout. Furthermore, if ATP depletion is suspected as a major cause, more statistics and a more concrete assessment of ATP depletion should be provided. This could be achieved for example by conducting characterization of vesicle sizes by cryoEM microscopy or related techniques to extract information on morphology and mean radii. Thus, the intraliposomal volume and ATP concentration can be estimated. Consequently, the time it would take to deplete ATP could be determined.

5. On page 4, lines 133-136: the authors provide a valid strategy to control for dye photobleaching. On the same rationale, I think representative donor-only and acceptor-only traces should be shown in the supplements to rule out dye blinking behaviors (again, especially at the presence of magnesium ions).

6. I found very interesting the paragraph on the characterization of the dissociation kinetics between S-components and ECF modules, using the MBP binding kinetics as fiducial standard. Particularly, how the authors elaborate on non-productive futile cycles of ATP hydrolysis happening within one dissociation cycle of the S-component. However, the use of indirect comparison with ATP hydrolysis turnover rates reported in literature to raise this claim and draw this conclusion left me skeptical. I think minimum requirements to sustain this hypothesis should be experiments where a bulk estimate of ATP hydrolysis turnover rate is validated and further compared with the dissociation rate of the S-component. The latter can be extracted by performing exponential fitting on the decay curves shown in Figure 4d.

7. What is the authors' take on the multiple intensity levels observed in both donor and acceptor channels (Figure 2a-2c), which suggest that the dynamics of more than one acceptor and donor molecule are monitored? For example, in Figure 2a the first acceptor trace shows two intensity steps.

Minor Comments

Smaller issues and concerns are also listed as follows:

1. To rule out FRET artifact due to non-specific binding of the dyes to detergent micelles or lipid environment, mock labeling experiments of cysteine-less ECF-CbrT complex at the presence of donor and acceptor dyes should also be provided.
2. I wondered whether the photobleaching lifetime of the fluorophores represented an obstacle which could be overcome for recording longer movies and pursuing the dwell-time analysis. For instance, I am curious to know whether the authors have attempted to characterize the longest observation time at which photobleaching is negligible by screening for different conditions (e.g. laser excitation, by introducing ROS scavengers and photo stabilizers, or introducing time lapses within the observation period).
3. I would like the authors to comment on the motivation behind the initial choice to perform confocal microscopy measurements in detergent micelles.
4. Page 5, line 152: I would like to know how the authors explain the significant percentage (19%) of observed static traces exhibiting low-FRET values.

Suggestions for Improvement

Last, I would like to propose a few adjustments, which in my opinion would help to improve the general structure and clarity of the manuscript:

- The authors nicely summarize the existence of two cobalamin translocation models postulated to describe the transport mechanism of ECF-CbrT. I think it would be highly beneficial for the flow of the paper, if the authors could state more clearly/less ambiguously in the discussion section whether their findings are more in favor of the first over the second model described in the introduction section, or viceversa.
- More clear figure sketches about design of different assay schemes to accompany what it is described in the text would help the flow. For example: graphic representation of the Transport cycle according to the two postulated models could be provided; the structures of H- and C- sensors should not be overlapped in Figure S2; I found a bit hard to link the two reconstitution schemes described on page 7 (lines 216-228) with Figure 3a. I think more graphic support is needed here.
- In the discussion section there is in my opinion too much emphasis on speculations and perspective in spite of a poor structure on re-wrapping the final interpretation of all the findings following the same order of the paragraphs of the results section.

Reviewer #3

(Remarks to the Author)

Reviewer #4

(Remarks to the Author)

In this study, the authors investigate the dynamics of the ECF-CbrT complex's association and dissociation under conditions mimicking the transport cycle. ECF transporters, which belong to the superfamily of ABC transporters, differ both structurally and mechanistically from canonical ABC transporters. The authors tackle a central question in the field: is the dynamic association and dissociation of the S-component from the ECF module part of the transport cycle?

To address this, the authors undertook the challenging task of applying single-molecule FRET (smFRET) in both detergent solution and reconstituted liposomes. This question lies at the heart of ECF transporter mechanism research, making the study important and timely. The approach chosen is highly suitable, and the potential impact of the work on the field is substantial. Overall, this is an excellent work, however, some technical issues need to be addressed prior to publication:

Main Concerns

While the study is promising, there are several points that need to be clarified or strengthened to enhance the authors' claims:

1. Linking Dynamics to Function:

In both the abstract and the main text, the authors emphasize that the core question is whether the complex remains stable throughout the transport cycle. Even if the authors' data support their interpretation of a dynamic complex, the link between these dynamics and transport function remains tenuous. It is not yet demonstrated whether the observed dynamics are necessary for transport. Are the dynamics an intrinsic part of the transport cycle, or can transport occur without cyclic association and dissociation? The manuscript would benefit from a clearer demonstration of this relationship.

2. Noise in Traces:

In Figures 1f and 1g, the donor and acceptor traces show significant noise, which is not correlated in time and does not appear to represent FRET events. Notably, this noise is much less apparent during the direct acceptor excitation period. This raises the question of whether the fluctuations observed could be due to real conformational dynamics occurring on a timescale faster than the experimental resolution. Can the authors exclude this possibility? A discussion of this point would be helpful.

3. Fluorescence Anisotropy:

It is essential that the fluorescence anisotropy (r) be measured and compared between the fluorophores in solution and those attached to the proteins. This data is missing, and its absence leaves open the possibility that restricted fluorophore orientations could introduce errors in distance measurements. I recommend that these measurements be included in the supplementary information.

4. Negative Control:

The inclusion of a negative control using an ATPase-deficient mutant is critical. This would help to solidify the interpretation of the ATP-dependent dynamics observed in the smFRET traces.

5. Magnesium and ATP Concentrations:

The use of 10 mM Mg-ATP in the experiments shown in Figure 2 and elsewhere raises concerns, as high magnesium concentrations (>2 mM) can disrupt liposomes, potentially confounding the experimental results. In light of the K_m -ATP of ~0.2 mM (according to Figure S5), 1 mM ATP should be sufficient for saturation. Along the same lines: was 10 mM magnesium also used with ADP and AMP-PNP? This detail is absent from the methods section and should be clarified.

6. Discrepancy in ATP Results:

In Figure 2, when ATP was added only to the exterior of the membrane, ~22% of traces displayed dynamic behavior. However, when ATP was added to both sides, the fraction of dynamic traces dropped to ~13%, despite the increase in the fraction of dissociated complexes (!). This discrepancy is difficult to reconcile, and the authors suggest it may stem from the ATP encapsulation procedure or ATP depletion during sample preparation. Both issues could be resolved by standardizing the preparation protocol and incorporating an ATP regeneration system.

7. FRET Efficiency Traces:

FRET efficiency traces are missing from Figures 2a and 2c. Without them, it is difficult for readers to correlate the time-dependent traces with the FRET distributions. Including these traces would greatly improve clarity.

8. Experiments with AMP-PNP:

Figure 2b suggests that the nucleotide-bound state (AMP-PNP) promotes a stable complex, while nucleotide hydrolysis is required for dissociation. Were the experiments in Figure 3 also conducted with AMP-PNP? If not, including these would strengthen the conclusions.

Additional Concerns

- The manuscript states that the experiments were repeated with saturating concentrations of vitamin B12. What is the K_d for B12 binding by the S-module? The use of 10 μ M seems rather high for a trace molecule like B12.
- The statement "it is very different from other ABC transporters, where ATPase activity is often stimulated by the transported substrate" should be revised to reflect the fact that only in some canonical ABC transporters is ATPase activity stimulated by the substrate, while in others, it is not.

• The manuscript mentions that "binding of maltose on MBP occurs around 10 s after the start of the pump, and is highly synchronized throughout the population (Fig. 4d), which is expected for a high-affinity binding protein." Why is this expected for a high-affinity binding protein? Are the authors referring to the fast onset or the homogeneity? In either case, high affinity typically correlates more with K_{off} than K_{on} , so a more detailed explanation would be helpful.

Version 1:

Reviewer comments:

Reviewer #1

(Remarks to the Author)

The authors have gone above and beyond to address in detail all my comments. They have provided solid statistical analysis, additional explanation and alluring and explanatory new figures.

The work was already stellar for the field; the extra work provided during the revision further solidified the important for the field findings. I highly recommend publication.

PS thanks to the authors for realizing I misread their text in my comment 7 (thought they wrote molar ratio (instead of w/w))

(Remarks on code availability)

Reviewer #2

(Remarks to the Author)

The authors performed extensive improvements and address constructively and thoroughly all the points and concerns raised during the revision process.

Focusing particularly on the reply to my section in the revision letter, the authors have conducted big progresses and integrated their work by introducing five new figures (between main and supplementary) and, where, specified, the text have been restructured, changes were made and incorporated accordingly. All this together helps to resolve many ambiguities and strengthens the content of this work.

However, unfortunately, I am reluctant to recommend publication because, in my opinion, this work does not represent a significant conceptual advancement for the field. The authors debated this issue in response to my point 1 of the previous revision round. It is true that this study provides a non-trivial methodological advancement for allowing conducting single-molecule investigation of these non-canonical ECF transporters in their native-like environment, unlike previous bulk methods. With their work, they are able to show that even in a vectorial native-like lipid environment CbrT dynamically associates with and dissociates from the ECF modules and ATP molecules likely induce association and dissociation events after undergoing multiple futile hydrolysis cycles. However, while this helps gaining confidence or corroborates what it has been proposed previously by studies conducted in artificial environment (such as DDM micelles or constrained lipid environment like nanodiscs), their findings do not seem to offer a significant conceptual advancement to our understanding of the transport mechanism of ECF transporters. For example and as it was also mentioned by Reviewer #4, correlating the substrate translocation to the association/dissociation dynamics could have undoubtedly made a big impact and strengthen the claim that this method offers a big advantage by conducting smFRET studies for the first time in a vectorial system. Or, as the authors also state in the discussion of the revised manuscript, conducting a more comprehensive quantitative analysis on the association/dissociation cycle by monitoring the dynamics over longer observation periods could have also strengthen significantly the value and application spectrum of this work.

Smaller points for which further clarity is recommended are:

1. In response to my point 5 of the revision, the authors have included a new Supplementary Figure 5 with a donut diagram, which show the proportions of the TIRF traces. I thank the authors for the effort, but I think more than only one trace should be presented for "No acceptor" and "No donor". In the latter case, two traces are shown but my impression is that one trace is "No donor AND No acceptor". I understand these are not the selected traces, but I also think that the donor-only and acceptor-only traces provide immediate confidence that no photophysical artifacts are occurring. I was particularly interested in "No donor" and "No acceptor" traces at the presence of ligands (MgATP and B12).

2. Supplementary Figure 9: how did the authors interpret that the dwell decays could be better fitted with a two-phase exponential is not clear to me. Are there a fast and a slow component in the binding/unbinding dynamics or is it sample heterogeneity?

3. My final comment in the revision process was to work on the structure of the discussion. I see the authors have worked on that and the structure improved a lot. However, I think that a more thorough refinement of the discussion should still be considered. Subjectively, I found the different parts in the discussion a bit fragmented and I had hard time trying to joint them together in a global picture.

(Remarks on code availability)

Reviewer #3

(Remarks to the Author)

(Remarks on code availability)

Reviewer #4

(Remarks to the Author)

Dear authors,

First, I would like to reiterate that this is a potentially interesting and important study, but still requires a few clarifications to fully support the conclusions.

Several studies (including ones performed by yourselves) had previously shown that for some ECF transporters the S-component and the ECF module dynamically associate/dissociate.

The potential novelty of this study would have been to show that these dynamics are an integral and essential part of the transport cycle, and to the best of my judgement, this has not been demonstrated.

In your response, you explain that performing simultaneous single molecule measurements of association/dissociation and of transport is a daunting task. I fully understand and accept this explanation.

However, there are other, simpler ways, to link the association/dissociation to function. One idea that comes to mind is to X-link the two proteins with linkers of various lengths and flexibility and measure at which point (if ever?!) transport is lost. If your hypothesis is correct, a short and rigid linker would impair transport, while a long and flexible one would not (as has been observed with LacY). This can readily be backed up by sm-FRET measurements, estimating the distance of separation that is required for transport. Maybe no separation, and hence no dissociation is required? For example, there are convincing studies that show a cyclic association-dissociation between the methionine transporter and its SBP (S-component equivalent). However, other equally convincing studies show that although these dynamics occur, they are perhaps not required for transport.

An alternative to the X-linking approach would be to create a chimera, fused at the genetic level. Similar experiments have been performed with other transport proteins to address such questions.

I FEEL THIS IS A CRITICAL POINT THAT REQUIRES RESOLVING

This concern is linked to other MAJOR CONCERNS raised by the data:

In the apo state, the complex is stable, with rare dissociation events. Under hydrolyzing conditions, some (low) percentage of the traces become dynamic, with insignificant difference between ATP (in) and ATP (both sides). To me this could mean only one of two things:

1- A significant fraction of the proteins is not active

2- The dynamics occur on a timescale faster than the temporal resolution of the measurements.

The confocal experiments argue against option 2, even though, if I understand correctly, these were performed in detergent solution, and the authors go in length explaining why such experiments fall short as they lack the membrane environment and directionality.

An additional related concern pertains to the lack of effect of AMP-PNP:

From a thermodynamic viewpoint, if the complex is stable in the apo form, and is "less" stable under hydrolyzing conditions, either the pre-hydrolysis or post-hydrolysis nucleotide bound state needs to be the less stable, dissociated state. In this respect, the data presents two troubling issues:

1- To the best of my knowledge, in "regular" bacterial ABC transporters that function as importers, ATP binding, as mimicked here by addition of AMP-PNP, is sufficient for formation of the closed NBDs dimer. Why would then AMP-PNP have no effect on association? From studies of other studies of ABC transporters (importers), the APO state represents one energy level, and the nucleotide bound one state another. Here, these two states seem functionally equivalent. This point is largely ignored by the authors.

2- If the authors think that the high-energy dissociated state is populated in the post hydrolysis state, this needs to be shown, and can readily be done so by trapping it.

MINOR COMMENTS

I believe the statistical analysis in 2C need to be done with ANOVA, not with an (two-tailed unpaired t-test

The subscript legends in 2b are too small to read and are also confusing. I imagine that "ATP" means ATP on both sides and ATPout means ATP only on the inside. This should be made clear in both the figure and legends.

"These values are close to the apparent KM value for ATP measured in bulk transport (190 μ M, Supplementary Fig. 8b), suggesting that the dissociation of the complex is the ATP dependent step in transport"

This sentence is quite confusing. Do the authors simply wish to say that these data suggest that indeed dissociation is

driven by ATP hydrolysis? If this is the case. I'd suggest saying just that.

“Unfortunately, the number of observed transitions within single movies was too low for such analysis, because of the slow transport kinetics (Fig. 2a,c)”

This claim is somewhat misleading since Figure 2A,C show association/dissociation dynamics and propensities and not transport.

(Remarks on code availability)

We thank the four reviewers for their extensive, constructive and very helpful comments on our manuscript, based on which we have performed additional experiments, included new and extended analyses, and rewrote the text in places where extra clarification was needed. We hope that you will now find the manuscript suitable for publication.

Reviewer #1 (Remarks to the Author)

Lefebvre et al present an informative and well designed work offering the direct observation of ATP-induced dynamics of vitamin B12 transporter ECF-CbrT under turnover conditions. The work primarily relies on single molecule FRET, confocal and camera based, as well as readout of activity for the transport in detergent and reconstituted in liposomes. Based on these readout authors find that the dynamics are dependent explicitly on ATP hydrolysis and are observed in the presence but also absence of vitamin B12. They also find that S-component expulsion and reassociation are an integral part of the translocation mechanism. The experimental work is nicely designed, the quality is sound, multiple control experiments are performed and the data support the authors claim in general. The work is stellar for the field, nicely written and in general easy to follow and important for a wide audience. Addressing the technical elements below will help authors to validate statistically the claims.

1. Authors conclude that ATP, and not vitamin B12, alters dissociation and association dynamics of the complex. Their conclusion is primarily relied on reporting minimal shift in FRET distribution and no change in the number of dynamic traces in the presence/absence of B12. There are a few important elements to be clarified for this central message to be carried out by the experiments:

a) The graph in 3d shows very similar % of dynamic/(non dynamic) in all 4 conditions of apo, B12, ATP and ATP+B12 ($11/14=0.78\%$, and $40/55=0.73\%$). This may be impaired by the very low statistics. Authors should comment on this

Response: We now realize that some confusion may have occurred because the data shown in Figure 2 and Figure 3d in the manuscript are not directly comparable. Figure 3d reports on the co-reconstitution experiment with singly labeled transporters. In this figure, we show the percentage of traces either displaying dynamics or static *high*-FRET. In contrast, in Figure 2 the non-dynamic traces comprised both static *high*- and static *low*-FRET. To illustrate the difference, we added a supplementary figure (now Supplementary Figure 7 in the revised manuscript and **Response Figure 1** below) to compare the data from the manuscript Figure 2 and 3 distinguishing between high-FRET, low-FRET and dynamic FRET. With the doubly labeled sensor, most traces show static high-FRET (associated) while in the co-reconstitution experiment the traces show mostly dissociated complexes. The static high-FRET population in the Figure 2 is from transporters that either never dissociated or dissociated and re-associated before the start of the recording. In Figure 3, only the latter will give static high-FRET as singly labeled transporters that never dissociated give rise to static low-FRET. The ratio between dynamic and static traces (the latter including both low and high FRET traces) for co-reconstituted singly labeled transporters in the conditions with ATP is around 1:4 (thus 20 % of the total is dynamic) which is similar to the doubly labeled transporters. Interestingly, among all traces displaying high-FRET (static plus dynamic), the percentage of dynamic traces is much higher in the co-reconstitution experiments than in the experiments with doubly labeled transporters. This could be because high-FRET in the co-reconstitution experiment will occur only if there is dynamics (the onset of the experiment is static low FRET), while with the doubly labeled transporters the onset is high-FRET. Finally, the reviewer

is correct that the number of dynamic and static high FRET traces in Figure 3 is lower than in Figure 2, inevitably affecting statistics. This is caused by the low chances of capturing two singly labeled transporters in the correct orientations in a single liposome in the co-reconstitution experiments of Figure 3.

Response Figure 1 | Distribution of traces in smTIRF recordings. Proportions of High-FRET, low-FRET and dynamic traces are shown for each condition. Co-reconstitution of single mutants is shown as in Figure 3 and the distributions for the double-labeled C-sensor are shown for comparison.

b) The figure 4d displays the decay with time of the % traces before transition. Authors claim that the B12+ATP trace is similar to the ATP only. However, the two decay appear not identical as authors write. To support this claim authors can provide a statistical test (either decay rates with exponential decays, or other appropriate models) and output a “t” test or KS test of their significance.

Response: To support the claim that the rate of first dissociation is similar in presence and absence of vitamin B12, we now included a one phase decay exponential fit on the percentage of traces before transition. Both the ATP-only condition and the ATP+B12 condition are significantly different from the maltose binding experiment using MBP ($P < 0.0001$, un-paired t-test) while not significantly different from each other ($P = 0.95$). Description and result of the fitting and statistical test, have been added to Figure 4 and its legend.

2. In the last paragraph of results authors write “that not every hydrolysis of ATP is coupled to

release of the S-component” and they justify this by comparing the turnover rate of 2-3 sec⁻¹ to the dwell time of transitions in fig2ac. The manuscript would greatly benefit if this is not qualitatively but quantitative. Authors in this case have enough traces to extract dwell times using HMM analysis of the SM FRET traces. they could use any of the existing tools/software that are for example in the recent work lead by S Schmidt and the FRET community in Nature commun 23

Response: We agree with the reviewer on the need for a quantitative comparison to strengthen the interpretation. We now included in the manuscript an ATPase activity assay in conditions comparable to the smFRET experiments (Supplementary Figure 8). At 23°C, a rate of hydrolysis of about 1 molecule of ATP every 2 seconds per transporter was recorded. We also did a quantitative analysis of the dwell times on the dynamic traces using the MASH FRET software¹. The analysis was done only on dwells of known duration, e.g. the first and last dwell time of each trace was not taken into account (as they last beyond the recording window). Consequently, static traces and traces with only one recorded transition were not considered at all. Two populations were found with a two-phase exponential decay fitting, giving dwell time half-lives of 5.8 - 17.8 s and 1.9 - 4.6 s (Supplementary Fig. 9). The first population has dwell times longer than the rate of ATP hydrolysis, while the second population has dwell time half-lives closer to the ATP hydrolysis rate. We stress that this dwell time analysis has a strong bias toward short dwell times because the dwells at the start and end of the recordings were not included and thus these half-lives are underestimations of the actual half-lives. Overall, these quantifications of dynamics rates are compatible with our interpretation that not every hydrolysis event is efficient in dissociating the S-component. For the comparison between dwell times and ATP hydrolysis rate, we also emphasize that for the former analysis the traces without dynamics, and those with only one transition were not used, while the proteins belonging to these classes are accounted for in the average ensemble of the ATPase assay, which also contributes to an underestimation of difference between the half-lives of the dwells and the ATP turnover rate. We added a description of the analysis in the methods section, and included dwell times description in the last paragraph of the results (line 311) and in the Supplementary figure 9.

3. Fig 1 confocal smfRET , why is the background in the 2D graphs so high ? it appears as noise is integrated., what are the thresholds utilized in the confocal data ?

Response: To identify bursts, we used the FRETbursts python package with the default search parameters (m=10, number of photons to compute photon rate and F=6 factor to multiply the background rate to get the minimum rate for a burst). Then, we applied a threshold to only consider the bursts with a minimum of 40 photons (during both excitation periods and in both channels). Increasing the threshold to 60 reduced the number of bursts by two without improvement of the background. We show below in **Response Figure 2b** bursts that are already selected for containing both donor and acceptor fluorophores, in which case the background is reduced. There, bursts are selected to have at least 25 photons during the donor excitation period (in acceptor and donor emission channels combined) and at least 15 in the acceptor channel during acceptor excitation periods. Both selection methods, with a minimum of 40 photons then selecting bursts within 0.25 to 0.85 stoichiometry or further selecting photon numbers in each excitation period, lead to a similar FRET efficiency distribution (**Response Figure 2d**).

As for noise, two potential sources are identified and could explain the larger distribution/higher background in confocal results. First, bursts where photobleaching happens are not selected out

(**Response Figure 2a**). Second, bursts where random coincidences happen (2 molecules at the same time in the confocal volume) are not selected out (**Response Figure 2c**). It is worth noting that confocal data distribution appears wider than the TIRF data because in the latter case bleaching molecules are excluded and time traces allow for noise reduction filtering.

Response Figure 2 | 2D distribution of C-Sensor bursts in Apo condition. a, FRET Efficiency / Stoichiometry (ES) plot of all bursts (threshold of minimum 40 photons), the green and red circles indicate the area where bursts would be located if donor or acceptor photobleaching occurs during the burst. **b**, ES plot of bursts containing both donor and acceptor (minimum 15 photons in acceptor channel during acceptor excitation period; minimum 25 photons during donor excitation period). **c**, Single cysteine mutant labeled separately and mixed together resulting in donor only and acceptor only bursts. The bursts with intermediate stoichiometry and low FRET efficiency correspond to random coincidences (two molecules in the confocal volume at the same time). **d**, FRET efficiency distribution of Bursts within the 0.25 - 0.85 stoichiometry range from (a) in black and from all the bursts from (b) in grey.

4. Authors should comment on how do they exclude potential variation in chromophore intensity readouts by the bilayer interaction and by the potential orientation of the ECF-CbrT complex. Do authors expect any effect of the orientation of the protein relative to liposomes to the actual dynamic reported here?

Response: We have previously determined that ECF-CbrT reconstitutes in both orientations in liposomes². However, based on the FRET efficiencies we do not appear to be able to distinguish between the two populations. Therefore, either the differently oriented complexes behave in the same way, or our measurements are not sensitive enough to observe differences. Variations in the dye intensity because of bilayer interactions cannot be ruled out, but Alexa Fluor 555 and Alexa Fluor 647 are among the best known dyes in terms of membrane interaction³. Nonetheless, we are careful not to rely on accurate distances between the fluorophores based on FRET levels. The co-reconstitution experiment shown in Figure 3 was done to show that dissociation and association indeed takes place, regardless of the exact distance between the fluorophores in the associated state. We now state explicitly in the manuscript why we found it important to conduct this experiment (line 237).

5. Labeling efficiency is ~68%. this would result in 3 out of 10 protein to be unlabeled. While this is easy to correct for intermolecular recording, it can affect the readout of the intermolecular FRET, at least the rates.

Response: In the experiments shown in figure 2, we select for traces with one donor and one acceptor dye. Any additional unlabeled protein complexes would need to have both cysteines unlabeled (because there are two cysteines in the complex) The percentage of transporters with both cysteines unlabeled is between 1 and 10, depending on the exact labeling efficiencies, which showed some batch-to-batch variation (68 to 95 %). We may occasionally have co-reconstituted two singly-labeled complexes (one with donor and the other with acceptor fluorophore) but the low protein-to-lipid ratio used in the experiments shown in Figure 2 reduces the chance of capturing two complexes in one liposome. The reviewer is right to point out that competition between labeled and unlabeled transporter might have an effect in the co-reconstitution experiment (Figure 3), artificially increasing the dwell times in low-FRET. Unfortunately, the number of dynamic traces from the co-reconstitution experiment is too limited to have a robust quantification of the dwell times in order to verify this hypothesis. Competition between modules is definitely an aspect of the ECF transporters function to explore further using smFRET in future research.

6. Authors report FRET efficiencies of 0.68 and 0.56 for the H-sensor and the C-sensor respectively, and compare with Ca distances of 35 Å. they could use the monte carlo simulation toolbox reported by Seidel group in Nat Methods 12, to accurately calculate from FRET efficiencies to Ca distances

Response: The reviewer is correct in stating that accurate conversion between FRET efficiency and distances would require more detailed analysis. We attempted to calculate the expected FRET efficiency using FRETpredict⁴ and the software suggested by the reviewer, FPS⁵. The results are shown below in **Response Figure 3a**. There is a difference of 0.2 to 0.3 FRET efficiency unit between the predictions but they are in general accordance with the experimental results. The difference can be explained by several approximations in the predictions such as the presence of the membrane/detergent micelle which are not taken into account and restrict the available volume of the dyes (**Response Figure 3b**). Additionally, the position of one cysteine on CbrT for the C-sensor is not structurally resolved and anisotropy was relatively high (see Supplementary Fig 3e for anisotropy and response to point 3 of reviewer #4 for further discussion). However, it is important to stress that, for the interpretation of our data, determination of the exact distances is

not needed. We interpret the high-FRET population as a complex between S-component and ECF module (regardless of the exact distance between the fluorophores), and the low-FRET population as a dissociated complex. We rephrased lines 126-139 in the manuscript to avoid the suggestion that we can accurately convert FRET values to distances.

Response Figure 3 | Prediction of the FRET efficiency. a, Results of the FRET efficiency predictions for the H- and C-sensor using the softwares FRETpredict and FPS. The results from FRETpredict are shown as in the case studies⁴, with the FRET efficiencies calculated using different averaging regimes, static (E_{static}) and dynamic ($E_{dynamic}$ and $E_{dynamic+}$) shown as histograms together with the average ($E_{average}$). The experimental values are shown as points for the confocal experiment in detergent and TIRF experiment in liposomes (Ex). The result from the FPS calculation is shown as the additional purple histogram bar. In both programs, the calculations have been made using Alexa Fluor 647 maleimide structural features for both donor and acceptor with a fixed R_0 of 51 Å. The residues where cysteines have been introduced are used as anchor point for the dyes, except for CbrT_A182C which is not resolved in the structure so the last resolved residue on the C-terminus has been used (175 and 174 in the case of FRETpredict as 175 didn't yield distance distribution results). **b,** Structure of ECF-CbrT with accessible volume clouds for both labeling positions in the two sensors used in the study. The CbrT and EcfA subunits on which the labels are attached are colored. As an indication, the approximate location of the lipidic membrane is shown as black lines (based on structural information from transporters embedded in nanodiscs structures⁶).

7. Authors write they used 1:10.000 (w/w) ratio to ensure 1 protein per liposome. A typical liposome of e.g 50nm contains 50000 lipids and if the above ratio is indeed the one used and if all protein is reconstituted, it would result to higher likelihood of containing a protein per liposome (actually more like 5). Authors should check and comment on that

Response: We think that the reviewer's calculation is not correct. To calculate the approximate number of transporters per liposomes we first estimated N, the number of lipids per liposomes with the formula : $N = ((4(d/2)^2 + (4(d/2 - h)^2)/a$ with the diameter of liposomes $d = 100$ nm, the thickness of the bilayer $h = 5$ nm and the area of a lipid head $a = 0.71$ Å². This gives around 80.000 lipids per liposomes. With a molecular weight of 120 kDa for the protein complex, and an average lipid molecular weight of 750 g/mol for lipid molecules, a 1:10.000 mass ratio (w:w) corresponds to 1 transporter per about 20 liposomes assuming a 100 % reconstitution efficiency.

8. The two FRET distributions recorded by confocal and TIRf have very different widths. Authors should comment and explain this

Response: This point is related to point 3 above. The factors contributing to the differences in widths are: 1) TIRF distributions are based on traces after CK filtering, which reduces the width.; 2) noise from photobleaching and random coincidence is specific to confocal measurements. In TIRF, we exclude from the analysis all the molecules that display donor or acceptor photobleaching; 3) The width of FRET distribution reflects noise in the data, as well as dynamics/heterogeneities of dyes' photophysics⁷ and protein structure, all averaged over the measured time-window. In TIRF experiments, the noise, and the dynamics of the proteins and dyes is averaged over 200 ms frame time, and everything faster than that does not contribute to the width of FRET distribution, while in confocal-based measurements, the averaging time is shorter, 1 - 10 ms, leading to broader FRET distributions; 4) Finally, instrumental noise from camera and background variations, are different between TIRF-based and confocal-based experiments, and affect the width of FRET-distributions.

9. None of the figures describes if data are from biological or technical replicates. Authors should provide statistical test to validate their conclusion for example in Figure 2E but also in other places. Also the number of technical and biological replicates is not clearly stated in some figures.

Response: We apologize for the omission. Supplementary table 3 has now been added to the manuscript with information about replicates. We also have included statistical tests, now reported in Figure 2, 4 and Supplementary Fig. 9.

Minor comments

10. The experiments with the maltose should be adequately described as if someone is not familiar with the type will not understand them, Information has been added to the text (Line 345), to the legend of Figure 4 and in Supplementary Figure 4.

L139 , sentence is unclear

We thank the reviewer for noticing this poorly phrased sentence. We now rephrased (line 156): "It indicates that the complex in the apo state remains stable when embedded in a lipid membrane where the protein can freely diffuse."

L62 deformation of membrane , are there references ?

References (15,16,18) have been added.

The article would benefit for an additional schematic representation (or a zoom with a colour code) where all the components are shown .

Schematics have been modified and added in figure 1,3 and Supplementary Fig 1 and 2f.

L215 , There seems like there is a double "the"

Corrected

The figure legend of figure 4 is unclear (especially for b and c) .
We have rephrased the legend and added information for clarity.

Reviewer #2 (Remarks to the Author)

Summary

I thank the authors for their work and effort, for pursuing the challenges of the study and for the time invested in compiling the manuscript. I appreciated this work, as it represents a rare extension of single-molecule FRET investigations to ABC transporters of less conventional architecture like ECF transporters and their unique membrane-embedded substrate binding (S-) components. The authors attempt to dissect mechanistic key aspects of the turnover cycle and its ligand dependencies with strategies based on various schemes of intra- and inter-molecular FRET. I found particularly elegant the rational of using half-labeled transporter complexes to directly reveal that the S-component can interact dynamically and switch between various ECF modules. Furthermore, this work attempts to elucidate ambiguous postulates of proposed models of transport mechanism for this specific transporter type. I think this acquires even more value considering that, as the authors highlight, one of the many therapeutic strategies proposed to fight pathogenic bacterial species and develop antimicrobials is based on targeting ECF transporters, due to their involvement in microbial virulence.

However, after carefully reading the manuscript, I identified a few points and concerns that I would like the authors to address and comment on before I can recommend publication. These points are listed below:

Major Comments

1. The authors emphasize that the one major obstacle towards the postulation of a unanimous transport model for ECF transporters is the lack of studies where vectorial translocation of cobalamin can be mimicked. Thus, the authors present for the first time single-molecule observations of type III ABC transporters (ECF) conformational dynamics conducted in compartments separated by a lipid bilayer, else proteoliposomes. The authors stress on the importance of having established a vectorial system for this specific transporter type. I would like the authors to elaborate on the advantages introduced by this novelty in the approach for studying the ECF transporters. This, particularly because my understanding is that the findings of this manuscript are in accordance to related methodologies previously employed by others to study the transport system of interest.

Response: The reviewer is correct that many of our findings are in accordance with experimental data acquired by different (ensemble) techniques. However, smFRET in liposomes makes it possible to obtain mechanistic insights that are not accessible in DDM solution or nanodisc reconstitution because the dynamics of subunit dissociation and association is very likely affected by the non-natural environment of the detergent micelle, and the constriction of the belt protein in nanodiscs. We now state these considerations explicitly in line 77. In addition, while our observations consolidate the model built from previous methodologies, our single molecule FRET experiments now also open the way to further quantitative analysis of kinetics, and for instance re-association of un-bound S-components (now stated from line 427).

2. In the second paragraph of the results section, the ATP role in association and dissociation

dynamics of S-component from ECF module is characterized. The authors observe about 22% of traces exhibiting FRET transitions at the presence of 10 mM ATP and 10mM Mg²⁺. They identify two causes to rationalize the observed low percentage of dynamic traces. While I find plausible the speculation about the low turnover rate impacting the occurrence of dynamic traces, I don't think the authors are working under conditions where ligands ATP and Magnesium ions are only selectively accessible to the sub-population of inside-out oriented transporters. Pillar biochemical studies conducted on ABC (and other) transporters (Liu et al., J Biol Chem, 1997; doi: 10.1074/jbc.272.35.21883) report that magnesium induces a lipid phase transition (Allen et al., Biochemistry, 1990; doi: 10.1021/bi00464a013) which causes liposomes permeabilization at concentration of 10 mM or higher. Furthermore, the working principle of the well-established NADH-coupled assay (<https://microbiology.ucdavis.edu/heyer/wordpress/wp-content/uploads/2013/11/ATPase-assay.pdf>) to assess spectrophotometrically the total ATP hydrolytic activity of transporters relies on the use of 10 mM Magnesium with the purpose to permeabilize lipid vesicles. I don't think the orientation of the transporter is one plausible cause to explain the low percentage of observed dynamic traces. I would like to understand the rational of the authors on this matter.

Response: we thank the reviewer for raising this point, because we were apparently not clear in our phrasing. Because we have 10 mM ATP and 10 mM Mg²⁺, most Mg²⁺ will be complexed with ATP. With a K_d of 50 μM, there will be only 0.7 mM free magnesium ions, which is not enough to permeabilize liposomes. We now state these considerations explicitly in line 169. In addition, our recent work on transport of vitamin B12 by ECF-CbrT in liposomes in ensemble measurements also shows that our proteoliposomes are tightly sealed in the presence of 10 mM Mg-ATP². Finally, also experiments with ADP and AMP-PNP were done using the same Mg²⁺ concentrations.

3. On a related note and to my knowledge, magnesium can introduce photophysical phenomena such as dye blinking at 10 mM (or higher) concentration. Therefore, it could represent a source of errors in the estimate of FRET efficiency and distance distribution. It would be ensuring if the authors could provide evidence, by which they are able to show that magnesium does not introduce artifacts in the observations (e.g., false FRET transitions). For example, this could be achieved by performing control experiments with dsDNA standard labeled with donor and acceptor pairs located at known distances, immobilized on surface and monitored at the presence and at the absence of magnesium ions. Alternatively, the apo state of ECF-CbrT at the presence and absence of magnesium (without ATP) could be recorded.

Response: Because we compare the traces in the presence of ATP with those in the presence of other nucleotides at the same Mg²⁺ concentration, we can rule out that our observations are based on an artefact from Mg²⁺ ions. We believe that there are enough internal controls in the manuscript. For instance, the experiments using ADP and AMP-PNP were done in the presence of the same Mg²⁺ concentration; in fact, the concentration of free Mg²⁺ is even slightly higher in the presence of ADP than in the presence of ATP, because ATP binds the ion more tightly. In addition, if our observations in the presence of ATP would be an Mg²⁺ artifact, we should see the same effect in the co-reconstitution intermolecular experiment (Figure 3d) in the presence of ATP where the overall distribution is different.

4. Page 6, line 182: The authors comment on the lower percentage of dynamic traces observed when ATP and Mg²⁺ are entrapped in the lumen of the liposomes. Their interpretation for this unexpected finding is that S-components spatially segregate from ECF modules, leading to vesicles containing multiple ECF or S-components. If that is true, it should be relatively easy to corroborate this interpretation with the quantification of traces containing more than one single molecule under these two conditions (vesicles with ATP_{out} and ATP_{out/in}).

Response: We thank the reviewer for this suggestion, and performed additional analysis on the traces. Proteoliposome preparation ahead of microscopy recording consists of freeze thawing cycles and extrusion through a polycarbonate filter in order to obtain unilamellar liposomes with a 100 nm diameter. In absence of ATP, we expect that a labeled ECF-CbrT complex will remain intact during this preparation step. However, when ATP is present in order to encapsulate it inside the liposomes, CbrT can be dissociated from the complex during this procedure, and chances are that ECF module and CbrT will end up in different liposomes, especially since we use a protein to lipid ratio of 1:10.000 (approximately 1 transporter for 20 liposomes). We thus expect to have in the un-selected data, more traces with only 1 fluorophore. We verified this hypothesis on a subset of data with either ATP present during the proteoliposomes preparation (ATP_{in/out}) or only present during the recording in the outside buffer (**Response Figure 4**). In this recording session, 12 % of traces with at least one donor molecule, contained only one donor (no acceptor and no multiple donors) in the ATP_{out} condition while this percentage went up to 21 % when ATP was present on both sides which is consistent with our hypothesis. In absence of ATP during the preparation, it is likely that a trace containing 1 donor and 1 acceptor is a double-label transporter while if S-component were dissociated during the preparation procedure, it is also possible that a trace containing 1 donor and 1 acceptor comes from a liposome containing two ECF modules, two S-components or one ECF module and one S-component with different orientations in the membrane. As a result, we expect that these combinations will give traces with no apparent FRET and decrease the percentage of dynamic traces. This section has been rephrased to clarify our interpretation (from line 198).

Response Figure 4 | Unselected traces with only one donor. Percentages of traces containing only 1 donor and no acceptors among traces with at least one donor with or without acceptors. Percentages were calculated for each movie from a recording session where both conditions ATP_{in/out} and ATP_{out} were recorded. Weighted average and standard deviations are calculated from

the number of traces with at least 1 donor in each movie (total number of traces are 2218 for ATP_{in/out} and 1162 for ATP_{out}).

The authors also debate that the futile cycles in the case of the “ATPout/in” condition due to ATP depletion can explain the observed lower dynamics. I don’t think this is a valid argument, because ultimately ATP is provided externally under similar conditions as in “ATPout”. Therefore, the percentage of dynamic traces should be at least equivalent to the dynamics found for ATPout.

Response: The reviewer is correct, as even in case of total inhibition by ADP in the lumen of liposomes, we would expect at least the level of dynamic traces seen in the ATP_{out} condition since fresh ATP solution is continuously pumped in the recorded channel of the flow-cell. Therefore, potential ADP inhibition cannot be a cause for less dynamics, and the spatial segregation (see point 4 above) is a more likely contributor to the observed difference. This part of the manuscript has been rephrased (from line 198). We emphasize that, even if we cannot completely explain the differences between the conditions with ATP on one side of the membrane (as in physiological conditions) and on both sides of the membrane (non-physiological), we can avoid the complications from the non-physiological condition by adding ATP only on one side of the membrane. We emphasize this consideration now in line 208.

Furthermore, if ATP depletion is suspected as a major cause, more statistics and a more concrete assessment of ATP depletion should be provided. This could be achieved for example by conducting characterization of vesicle sizes by cryoEM microscopy or related techniques to extract information on morphology and mean radii. Thus, the intraliposomal volume and ATP concentration can be estimated. Consequently, the time it would take to deplete ATP could be determined.

Response: As discussed in the previous two points, we rephrased our interpretation about the cause of the difference between the ATP in and out and ATP_{out} conditions. ATP depletion has possibly a minor role in the experiment. This might be tested in the future with an ATP regeneration system⁸. For now, it is important that we can simply avoid complications caused by having ATP on both sides of the membrane.

5. On page 4, lines 133-136: the authors provide a valid strategy to control for dye photobleaching. On the same rational, I think representative donor-only and acceptor-only traces should be shown in the supplements to rule out dye blinking behaviors (again, especially at the presence of magnesium ions).

Response: Such traces have been included in Supplementary Figure 5.

6. I found very interesting the paragraph on the characterization of the dissociation kinetics between S-components and ECF modules, using the MBP binding kinetics as fiducial standard. Particularly, how the authors elaborate on non-productive futile cycles of ATP hydrolysis happening within one dissociation cycle of the S-component. However, the use of indirect comparison with ATP hydrolysis turnover rates reported in literature to raise this claim and draw this conclusion left me skeptical. I think minimum requirements to sustain this hypothesis should be experiments where a bulk estimate of ATP hydrolysis turnover rate is validated and further

compared with the dissociation rate of the S-component. The latter can be extracted by performing exponential fitting on the decay curves shown in Figure 4d.

Response: We agree with the reviewer (similar to point 2, reviewer #1) that a quantitative comparison is needed for the interpretation. We now included in the manuscript an ATPase activity assay (Supplementary Figure 8) and a quantitative analysis of the dwell times on the dynamic traces was done using the MASH FRET software¹ (Supplementary Figure 9). Then quantitative analysis of dynamics rates are consistent with our interpretation that not every hydrolysis event is efficient in dissociating the S-component. See response to reviewer #1 for a more elaborate discussion.

7. What is the authors' take on the multiple intensity levels observed in both donor and acceptor channels (Figure 2a-2c), which suggest that the dynamics of more than one acceptor and donor molecule are monitored? For example, in Figure 2a the first acceptor trace shows two intensity steps.

Response: We apologize for the confusion caused. If we understand correctly, the reviewer interprets the decrease in acceptor intensity (from 30.000 a.u. to 12.000 a.u.) in the first trace of Fig. 2a as a bleaching step. This is indeed how data would look like in experiments when protein copy number is determined via counting of photobleaching steps. However, in our experiments, the first 5 s of the recordings are with direct excitation of the acceptor, and the first decrease is due to a switch from direct excitation with a red laser to FRET-sensitized excitation with a green laser. The next drop in acceptor intensity (around 38 seconds in the first trace of figure 2a) is the switch from the high-FRET to the low-FRET state that we are interested in. At both the start and the end of every recording, we directly excite the acceptor for 5 seconds (pink shaded areas), just to make sure that we do not lose the acceptor fluorophore from bleaching. This point is emphasized in lines 149.

Minor Comments

1. To rule out FRET artifact due to non-specific binding of the dyes to detergent micelles or lipid environment, mock labeling experiments of cysteine-less ECF-CbrT complex at the presence of donor and acceptor dyes should also be provided.

Response: Our data doesn't show any evidence of such an artifact: in gel fluorescence shows specific labeling of cysteine containing subunits (EcfA and CbrT while no fluorescence is visible in EcfA' and EcfT bands), and the degree of labeling is always below 100 % meaning the total concentration of dye measured (donor + acceptor) is less than twice the protein concentration measured after correction for dyes absorbance at 280 nm. In addition, the fluorophores used in this study have been shown to be among the least interacting with the membrane³.

2. I wondered whether the photobleaching lifetime of the fluorophores represented an obstacle which could be overcome for recording longer movies and pursuing the dwell-time analysis. For instance, I am curious to know whether the authors have attempted to characterize the longest observation time at which photobleaching is negligible by screening for different conditions (e.g. laser excitation, by introducing ROS scavengers and photo stabilizers, or introducing time lapses within the observation period).

Response: Indeed, parameters for the recording procedure have been optimized in the preliminary stage of this work. Laser power, frame rate and movie length have been tested to optimize the signal to noise ratio, the time resolution and the photobleaching rate. Longer movies resulted in very few traces with donor and acceptor without photobleaching. Oxygen scavenger (PCA/PCD) and photostabilizer (Trolox) are present in the recording buffer (as mentioned in the methods section). We have not tested time lapses, it would be interesting to use it in the future, to investigate whether individual transporters switch between different kinetics of association/dissociation.

3. I would like the authors to comment on the motivation behind the initial choice to perform confocal microscopy measurements in detergent micelles.

Response: We apologize for the unclarity on the reason why we did the confocal experiments. The confocal experiments were a fast way to characterize the large number of mutants that we designed to develop the FRET sensors, on a set-up routinely used in the lab⁹, before going to the optimization in liposomes. The TIRF setup was “new” and recording and analysis procedures needed to be optimized, so confocal data allowed us to get a crosscheck between the two microscopy techniques and we thought it was important to show this data to demonstrate the robustness of our approach. This is now highlighted in the manuscript (line 121).

4. Page 5, line 152: I would like to know how the authors explain the significant percentage (19%) of observed static traces exhibiting low-FRET values.

Response: As we show in Figure 2, some dynamic traces contain only one FRET transition indicating that the association/dissociation process can be slow. If a dissociation took place before recording started (for instance in the 5 seconds of direct acceptor excitation) It is possible that dissociated complexes take longer than the length of the recording to re-associate, resulting in apparent static low-FRET trace. Of note, such traces are also seen in absence of ATP but in smaller proportions (see distribution plots in figure 2 of the manuscript and **Response Figure 1**). It is possible that a small fraction of those traces corresponds to inactive transporters or conditions where no high-FRET can be seen (e.g. co-localization of liposomes containing singly-labeled transporters, or co-reconstituted singly-labeled transporter with the label on the same subunit).

Suggestions for Improvement

Last, I would like to propose a few adjustments, which in my opinion would help to improve the general structure and clarity of the manuscript:

- The authors nicely summarize the existence of two cobalamin translocation models postulated to describe the transport mechanism of ECF-CbrT. I think it would be highly beneficial for the flow of the paper, if the authors could state more clearly/less ambiguously in the discussion section whether their findings are more in favor of the first over the second model described in the introduction section, or viceversa.

Response: We have now modified the discussion section to more clearly state that the first model (now shown schematically in Supplementary Fig 1a) can better explain our findings than the second model.

- More clear figure sketches about design of different assay schemes to accompany what it is described in the text would help the flow. For example: graphic representation of the Transport cycle according to the two postulated models could be provided; the structures of H- and C-sensors should not be overlapped in Figure S2; I found a bit hard to link the two reconstitution schemes described on page 7 (lines 216-228) with Figure 3a. I think more graphic support is needed here.

Response: We modified and added schematic representations of the experimental design in Figures 1 and 3, Supplementary Figure 1,2, 3 and 4.

- In the discussion section there is in my opinion too much emphasis on speculations and perspective in spite of a poor structure on re-wrapping the final interpretation of all the findings following the same order of the paragraphs of the results section.

Response: We have followed the suggestion of the reviewer and now start the discussion section with a re-wrapping, before discussing perspectives.

Reviewer #4 (Remarks to the Author):

In this study, the authors investigate the dynamics of the ECF-CbrT complex's association and dissociation under conditions mimicking the transport cycle. ECF transporters, which belong to the superfamily of ABC transporters, differ both structurally and mechanistically from canonical ABC transporters. The authors tackle a central question in the field: is the dynamic association and dissociation of the S-component from the ECF module part of the transport cycle?

To address this, the authors undertook the challenging task of applying single-molecule FRET (smFRET) in both detergent solution and reconstituted liposomes. This question lies at the heart of ECF transporter mechanism research, making the study important and timely. The approach chosen is highly suitable, and the potential impact of the work on the field is substantial. Overall, this is an excellent work, however, some technical issues need to be addressed prior to publication:

Main Concerns

While the study is promising, there are several points that need to be clarified or strengthened to enhance the authors' claims:

1. Linking Dynamics to Function:

In both the abstract and the main text, the authors emphasize that the core question is whether the complex remains stable throughout the transport cycle. Even if the authors' data support their interpretation of a dynamic complex, the link between these dynamics and transport function remains tenuous. It is not yet demonstrated whether the observed dynamics are necessary for transport. Are the dynamics an intrinsic part of the transport cycle, or can transport occur without cyclic association and dissociation? The manuscript would benefit from a clearer demonstration of this relationship.

Response: We agree with the reviewer that the definitive answer to this question has not been given yet. The next step to do so is to combine the measurements on association and dissociation

dynamics (presented here) with measurement of transport at the single molecule level, by using a sensor for vitamin B12 (as done in bulk²). Such simultaneous detection of conformational dynamics and transport requires an upgrade of our current TIRF setup, so that more colors can be imaged. We now explain this point explicitly in the manuscript (from line 400). We would like to emphasize that -while not formally conclusive- the kinetics of dissociation and association observed here, and the turnover numbers for transport of vitamin B12 are consistent with the notion that transport and dynamics are linked.

2. Noise in Traces:

In Figures 1f and 1g, the donor and acceptor traces show significant noise, which is not correlated in time and does not appear to represent FRET events. Notably, this noise is much less apparent during the direct acceptor excitation period. This raises the question of whether the fluctuations observed could be due to real conformational dynamics occurring on a timescale faster than the experimental resolution. Can the authors exclude this possibility? A discussion of this point would be helpful.

Response: Fast dynamics within long-lived states cannot be excluded based on TIRF data, and in confocal data we see wider FRET distributions that can be interpreted this way, but also in several other ways (see Response to point 8 of reviewer #1). Using confocal experiments for which the time resolution is much higher than for TIRF, we did photon by photon analysis¹⁰ and did not find indication of multiple FRET states. Background from lipids might be higher in donor than in acceptor channel, which gives rise to different noise levels. Additionally, besides conformational dynamics and FRET changes, the noise upon FRET-synthesized acceptor excitation can be higher than upon direct acceptor excitation due to donor photoblinking.

3. Fluorescence Anisotropy:

It is essential that the fluorescence anisotropy (r) be measured and compared between the fluorophores in solution and those attached to the proteins. This data is missing, and its absence leaves open the possibility that restricted fluorophore orientations could introduce errors in distance measurements. I recommend that these measurements be included in the supplementary information.

Response: We have done the fluorescence anisotropy measurements, and - consistent with literature data⁹ - the anisotropy is relatively high for the fluorophores that we used (Alexa Fluor 555 and Alexa Fluor 647). Therefore, indeed distance determination should be performed with caution. However, for the interpretation of our data, determination of the exact distances is not needed (see also point 6 of reviewer #1). We interpret the high-FRET population as a complex between S-component and ECF module (regardless of the exact distance between the fluorophores), and the low-FRET population as dissociated complex. The experiments presented in Figure 3 were done to support this interpretation. We have now mentioned this point more explicitly in line 237. Because the trigger for dissociation is the addition of Mg-ATP, it is also important to note that the anisotropy is not affected by the presence of Mg-ATP (Supplementary Figure 3).

4. Negative Control:

The inclusion of a negative control using an ATPase-deficient mutant is critical. This would help to solidify the interpretation of the ATP-dependent dynamics observed in the smFRET traces.

Response: We agree with the reviewer that such a control would strengthen our interpretation and had already attempted mutation of the Walker B glutamates (which are commonly mutated to inactivate ATPases of ABC transporters) on the cysteine-less background. Unfortunately, the mutated complex aggregated (**Response Figure 5a**), which made it impossible to use as ATPase inactive control. The aggregation is probably caused by the closed conformation that the ATPases have in the mutant, which leads to expulsion of the S-component¹¹. Therefore, it is impossible to purify the four-subunit complex with mutated Walker B glutamates. Thus, we decided to only use Mg-ADP and Mg-AMP-PNP as negative control for ATP hydrolysis. It is noteworthy that mutation of the catalytic glutamates in the walker B motifs of the ATPases has been recently described in the ECF module (without S-component) in a publication from our lab (Methods section and supplementary figure 7a from Thangaratnarajah, C. et al. Nat. Comm 2023 <https://doi.org/10.1038/s41467-023-40266-1>)¹¹. Also in that case, substitution of the glutamates to glutamines lead to an prominent aggregation peak in size exclusion chromatography with a change in the stoichiometry with EcfA more present as compared to EcfA' (**Response Figure 5b**). This issue was overcome in the case of the ECF module by expressing EcfA' in the second open reading frame (orf) from the p2bad plasmid. This solution cannot be applied in our case as this orf is occupied by the S-component CbrT.

Response Figure 5 | ECF-CbrT inactive ATPase mutant. **a**, Size exclusion chromatography profile of the ATP hydrolysis deficient ECF-CbrT mutant on the cysteine-less background (EcfA_E171A, EcfA'_C168A_E171A, EcfT_C252S, CbrT_C22S_C86S) showing a prominent aggregation peak (1) overlapping with the expected elution volume (2). **b**, SDS-PAGE gel of two collected fractions from the size exclusion illustrating the change in stoichiometry with a thicker EcfA band, the corresponding fractions are highlighted in panel a.

5. Magnesium and ATP Concentrations:

The use of 10 mM Mg-ATP in the experiments shown in Figure 2 and elsewhere raises concerns, as high magnesium concentrations (>2 mM) can disrupt liposomes, potentially confounding the experimental results. In light of the K_m -ATP of ~0.2 mM (according to Figure S5), 1 mM ATP should be sufficient for saturation. Along the same lines: was 10 mM magnesium also used with ADP and AMP-PNP? This detail is absent from the methods section and should be clarified.

Response: This point is similar to point 2 from reviewer #2. Again, we thank the reviewer for raising this point, because our phrasing was ambiguous. Because we have 10 mM ATP and 10 mM Mg^{2+} , most Mg^{2+} will be complexed with ATP. With a K_D of 50 μ M, there will be only 0.7 mM free magnesium ions, which is not enough to permeabilize liposomes. We now state these considerations explicitly from lines 169. In addition, our recent work on transport of vitamin B12 by ECF-CbrT in liposomes in ensemble measurements also shows that our proteoliposomes are tightly sealed in the presence of 10 mM Mg-ATP². Finally, also experiments with ADP and AMP-PNP were done using the same Mg^{2+} concentrations.

6. Discrepancy in ATP Results:

In Figure 2, when ATP was added only to the exterior of the membrane, ~22% of traces displayed dynamic behavior. However, when ATP was added to both sides, the fraction of dynamic traces dropped to ~13%, despite the increase in the fraction of dissociated complexes (!). This discrepancy is difficult to reconcile, and the authors suggest it may stem from the ATP encapsulation procedure or ATP depletion during sample preparation. Both issues could be resolved by standardizing the preparation protocol and incorporating an ATP regeneration system.

Response: We now have corroborated our hypothesis that the ATP encapsulation procedure contributes to the reduced number of dynamic traces (see point 4 from reviewer #2, and **Response Figure 2**). We also agree with reviewer #2 that ATP depletion is unlikely to be the cause of the reduction in dynamic traces. We feel that we may have emphasized the discrepancy too much in the manuscript (see also response to reviewer #2). Instead of getting to the bottom of the apparent discrepancy between the conditions with ATP on one side of the membrane (as in physiological conditions) and on both sides of the membrane (non-physiological), we decided to avoid the possible complications from the non-physiological condition (with ATP on both sides requiring ATP during the liposomes preparation) by only using conditions in which ATP is present on one side of the membrane. In this way, we used a physiologically relevant, robust, and consistent experimental design throughout the manuscript. We emphasize this consideration now in line 208.

7. FRET Efficiency Traces:

FRET efficiency traces are missing from Figures 2a and 2c. Without them, it is difficult for readers to correlate the time-dependent traces with the FRET distributions. Including these traces would greatly improve clarity.

Response: FRET efficiency traces have been added to Figure 2 and 3 in addition to the donor and acceptor intensity traces.

8. Experiments with AMP-PNP:

Figure 2b suggests that the nucleotide-bound state (AMP-PNP) promotes a stable complex, while nucleotide hydrolysis is required for dissociation. Were the experiments in Figure 3 also conducted with AMP-PNP? If not, including these would strengthen the conclusions.

Response: We certainly considered the suggested experiment as an extra control, besides the apo-condition. However, because of the labour intensiveness of the TIRF experiments, and our acquired insight into the functioning of AMP-PNP on ECF transporters from past experiments (e.g. it does not induce closure of the ATPase dimer¹¹), we decided on using the apo-conditions as controls.

Additional Concerns

- The manuscript states that the experiments were repeated with saturating concentrations of vitamin B12. What is the K_d for B12 binding by the S-module? The use of 10 μM seems rather high for a trace molecule like B12.

Response: Indeed, the vitamin B12 concentration is well above the K_d of 9.2 nM which was measured in a previous publication from the group¹². However, to ensure that several molecules of vitamin B12 are encapsulated in each liposome, we needed a concentration of at least 10 μM .

- The statement "it is very different from other ABC transporters, where ATPase activity is often stimulated by the transported substrate" should be revised to reflect the fact that only in some canonical ABC transporters is ATPase activity stimulated by the substrate, while in others, it is not.

Response: The sentence was removed from the manuscript and this statement has been rephrased (line 383).

- The manuscript mentions that "binding of maltose on MBP occurs around 10 s after the start of the pump, and is highly synchronized throughout the population (Fig. 4d), which is expected for a high-affinity binding protein." Why is this expected for a high-affinity binding protein? Are the authors referring to the fast onset or the homogeneity? In either case, high affinity typically correlates more with K_{off} than K_{on} , so a more detailed explanation would be helpful.

Response: Fast k_{on} ($2.3 \times 10^7 \text{ M}^{-1} \text{ s}^{-1}$) and homogeneity in the timescales (200 ms) used for averaging in TIRF experiment ($k_{off} \text{ } 90 \text{ s}^{-1}$) have been reported for maltose binding to MBP ($K_D = 1.2 \times 10^{-6} \text{ M}$)¹³ which makes it a good positive control.

Response References

1. Börner, R. *et al.* Simulations of camera-based single-molecule fluorescence experiments. *PLoS One* 13, e0195277 (2018).
2. Nijland, M., Lefebvre, S. N., Thangaratnarajah, C. & Slotboom, D. J. Bidirectional ATP-driven transport of cobalamin by the mycobacterial ABC transporter BacA. *Nat Commun* 15, 1–9 (2024).
3. Hughes, L. D., Rawle, R. J. & Boxer, S. G. Choose Your Label Wisely: Water-Soluble Fluorophores Often Interact with Lipid Bilayers. *PLoS One* 9, e87649 (2014).
4. Montepietra, D., Tesei, G., Martins, M. & Kunze, M. B. A. FRETpredict: A Python package for FRET efficiency predictions using rotamer libraries. 1–39 (2023).

5. Kalinin, S. *et al.* A toolkit and benchmark study for FRET-restrained high-precision structural modeling. *Nat Methods* 9, 1218–1225 (2012).
6. Thangaratnarajah, C., Rheinberger, J., Paulino, C. & Slotboom, D. J. Insights into the bilayer-mediated toppling mechanism of a folate-specific ECF transporter by cryo-EM. *Proceedings of the National Academy of Sciences* 118, e2105014118 (2021).
7. Kalinin, S., Sisamak, E., Magennis, S. W., Felekyan, S. & Seidel, C. A. M. On the origin of broadening of single-molecule FRET efficiency distributions beyond shot noise limits. *Journal of Physical Chemistry B* 114, 6197–6206 (2010).
8. Heinen, L., van den Noort, M., King, M. S., Kunji, E. R. S. & Poolman, B. Synthetic syntrophy for adenine nucleotide cross-feeding between metabolically active nanoreactors. *Nat Nanotechnol* (2024) doi:10.1038/s41565-024-01811-1.
9. Noort, M. van den, Drougkas, P., Paulino, C. & Poolman, B. The substrate-binding domains of the osmoregulatory ABC importer OpuA physically interact. *Elife* 12, RP90996 (2023).
10. Harris, P. D. *et al.* Multi-parameter photon-by-photon hidden Markov modeling. *Nat Commun* 13, 1–12 (2022).
11. Thangaratnarajah, C. *et al.* Expulsion mechanism of the substrate-translocating subunit in ECF transporters. *Nat Commun* 14, 4484 (2023).
12. Santos, J. A. *et al.* Functional and structural characterization of an ECF-type ABC transporter for vitamin B12. *Elife* 7, 1–16 (2018).
13. Telmer, P. G. & Shilton, B. H. Insights into the Conformational Equilibria of Maltose-binding Protein by Analysis of High Affinity Mutants. *Journal of Biological Chemistry* 278, 34555–34567 (2003).

Reviewer #1 (Remarks to the Author):

The authors have gone above and beyond to address in detail all my comments. They have provided solid statistical analysis, additional explanation and alluring and explanatory new figures.

The work was already stellar for the field; the extra work provided during the revision further solidified the important for the field findings . I highly recommend publication.

PS thanks to the authors for realizing I misread their text in my comment 7 (thought they wrote molar ratio (instead of w/w)

>We thank the reviewer for recognizing the value of our work. This has been a colossal effort, with the lead author spending 4 years -including the corona period (Jan 2020-May 2024)- to develop the smFRET assays.

Reviewer #2 (Remarks to the Author):

The authors performed extensive improvements and address constructively and thoroughly all the points and concerns raised during the revision process. Focusing particularly on the reply to my section in the revision letter, the authors have conducted big progresses and integrated their work by introducing five new figures (between main and supplementary) and, where, specified, the text have been restructured, changes were made and incorporated accordingly. All this together helps to resolve many ambiguities and strengthens the content of this work. However, unfortunately, I am reluctant to recommend publication because, in my opinion, this work does not represent a significant conceptual advancement for the field. The authors debated this issue in response to my point 1 of the previous revision round. It is true that this study provides a non-trivial methodological advancement for allowing conducting single-molecule investigation of these non-canonical ECF transporters in their native-like environment, unlike previous bulk methods. With their work, they are able to show that even in a vectorial native-like lipid environment CbrT dynamically associates with and dissociates from the ECF modules and ATP molecules likely induce association and dissociation events after undergoing multiple futile hydrolysis cycles. However, while this helps gaining confidence or corroborates what it has been proposed previously by studies conducted in artificial environment (such as DDM micelles or constrained lipid environment like nanodiscs), their findings do not seem to offer a significant conceptual advancement to our understanding of the transport mechanism of ECF transporters. For example and as it was also mentioned by Reviewer #4, correlating the substrate translocation to the association/dissociation dynamics could have undoubtedly made a big impact and strengthen the claim that this method offers a big advantage by conducting smFRET studies for the first time in a vectorial system. Or, as the authors also state in the discussion of the revised manuscript, conducting a more comprehensive quantitative analysis on the association/dissociation cycle by monitoring the dynamics over longer observation periods could have also strengthen significantly the value and application spectrum of this work.

> We thank the reviewer for the positive evaluation of the strength of our work, but we surprised by the perceived lack of conceptual advance. We will explain by distinguishing between the claims of the manuscript and the experiments asked for.

Claims in the manuscript: All reviewers unanimously agree that our experiments conclusively support the main claim of the manuscript which is captured in the title “Single-molecule visualization of ATP-induced dynamics of the subunit composition of an ECF transporter complex under turnover conditions”. They also agree that our development of an smFRET workflow in liposomes under turnover condition is a major achievement. Finally, they support our quantitative analysis showing that ATP hydrolysis occurs at faster timescales than the association-dissociation events, which is important for mechanistic interpretation, and is impossible to assess by methods different than single molecule experiments. This is a tremendous advance that in our view should warrant publication. This is an editorial decision to make.

The reviewer notion that there is not enough “conceptual advance”, is difficult to reconcile with the advances described above. In this particular case, there is an extra dimension that the reviewers may not be aware of: previous experiments showing that ECF transporter complexes can dissociate have been heavily criticized, because they were done largely with detergent-solubilized transporters, which were considered artifactual (see for instance the reviewer comments that we received in *Nature Communications* **volume 14**, Article number: 4484 (2023)). This is why we embarked on our mammoth effort using smFRET to study the expulsion using liposome-reconstituted transporters. There could have been two outcomes of: either the previously criticized detergent-based experiments were artifacts indeed, or they withstand the scrutiny of experimental testing using the more physiologically relevant liposome system. We find the latter, which is hugely important, also as standard for the field – even if it may seem that it had all been done already.

Experiments requested by reviewer#4: The experiments that reviewer #4 wants us to do relate to a claim that we *do not make* in the manuscript, namely that the movement of the substrate molecule across the membrane is strictly dependent on the association-dissociation dynamics of the complex. Although our data strongly suggests such a relationship, we do not claim it exists. For instance, in the discussion we write: “While the results show that dissociation of the S-component takes place in an ATP-dependent way, it does not formally show that expulsion of the S-component is essential for transport to occur.... However, although not formally conclusive, the kinetics of dissociation and association observed here, and the turnover numbers for transport of vitamin B12 determined from ensemble experiments (~0.01 per second) are consistent with the notion that transport and dissociation-association dynamics are linked.”

Obviously, a next step will be to investigate the potential mechanistic coupling (as we state in the manuscript), but this is way beyond the scope of the current manuscript. The experiments suggested by reviewer #4 would take at least another year to

conduct (on top of the four years of work in the current manuscript) and -based on similar experiments that we and others have done in the past- are at best going to be inconclusive, as has been shown repeatedly in the past based on similar fusion and crosslinking experiments. We therefore find the suggested experiments not only out of scope of the current manuscript, but also inappropriate.

Smaller points for which further clarity is recommended are:

1. In response to my point 5 of the revision, the authors have included a new Supplementary Figure 5 with a donut diagram, which show the proportions of the TIRF traces. I thank the authors for the effort, but I think more than only one trace should be presented for “No acceptor” and “No donor”. In the latter case, two traces are shown but my impression is that one trace is “No donor AND No acceptor”. I understand these are not the selected traces, but I also think that the donor-only and acceptor-only traces provide immediate confidence that no photophysical artifacts are occurring. I was particularly interested in “No donor” and “No acceptor” traces at the presence of ligands (MgATP and B12).

> The reviewer is right that the top trace in the no-donor panel also seems to have no acceptor. To further rule out the possibility of photophysical artefacts due to the recording condition in presence of ATP and vitamin B12, we show additional donor-only and acceptor-only traces in the recording condition (Figure Response 1 also added to the manuscript as Supplementary Figure 9). We now mention this additional control in the manuscript line 293. In Response Figure 1a we display selected traces (from one recording session) containing one donor and one acceptor. The distribution of donor intensities (top panel) and FRET efficiencies (middle panel) are shown overtime and one representative trace is shown (lower panel). Two FRET states are observed. The same representation is done in Response figure 1b but with traces that contained only one donor dye and no acceptor. To verify that ATP and vitamin B12 have no effect on the acceptor dye, we looked at the data recorded from the single cysteine mutant labeled with Alexa Fluor 647 that was used for the co-immobilization experiment (Supplementary Figure 6). This sample was recorded with direct acceptor excitation during 90 s to determine the fluorescence intensity corresponding to 1 acceptor. In presence of ATP and vitamin B12, the fluorescence intensity of the acceptor is homogenous.

Sample	C-Sensor doubly labeled ratio 1:10.000		EcfA_K122C_AlexaFluor647 ratio 1:10.000
Condition	10 μ M B12 in 10 mM ATP out		
Recording	7 movies; 3310 extracted traces from green channel		6 movies; 1449 extracted traces from red channel
# Traces	98	415	304

Response Figure 1 | Overall distribution of FRET efficiencies and intensities in turnover conditions. For traces containing 1 donor and 1 acceptor (**a**); only 1 donor (**b**) and only 1 acceptor (**c**), 2D histograms in top and middle panels show the distribution of fluorescence intensity and FRET efficiency values of the traces from one recording session (number of traces indicated in the table above). Bottom panels show one representative trace for each condition. Purple shaded areas indicate direct acceptor excitation in the first and last 5 seconds for the donor containing traces while for the acceptor only trace, the first 5 seconds (green shaded area) correspond to donor excitation.

2. Supplementary Figure 9: how did the authors interpret that the dwell decays could be better fitted with a two-phase exponential is not clear to me. Are there a fast and a slow component in the binding/unbinding dynamics or is it sample heterogeneity?

We initially showed two-phase exponential because it is a better model according to the BIC score. The reviewer is right to suggest that several causes could lead to this observation, fast/slow dynamic properties of the transporter, sample heterogeneity or it could be caused by the bias discussed in the manuscript of taking only the dwells that are fully captured during the recording time (dwells longer than the recording are not considered). At this point it would be difficult to explain the observation that two-phase exponential is a better model without overinterpreting our results. Thus, we now show one-phase exponential fittings in Response Figure 2 (and in the Supplementary Figure which is now number 10). In addition, we corrected a mislabeling in Supplementary Figure 10a. The important information we take away from this dwell time analysis, is that overall, the dwell times are longer than the hydrolysis rate which supports our interpretation that not every hydrolysis event is efficient in dissociating the S-component. Apparently, for every dissociation event of the S-component, several unsuccessful attempts take place.

Response Figure 2 | Dwell time analysis of the associated and dissociated transporter.

a, Distribution of dwell times of associated (High-FRET) or dissociated (Low-FRET) complexes in either ATP_{out} (blue) or ATP_{out} + B12_{in} (red) conditions. One-phase exponential decay fits are shown as dashed lines for each condition **b**, Table recapitulating the number of traces analyzed and dwell times extracted. Parameters of the exponential fitting are shown A is the scaling factor, τ the mean dwell time and t_{1/2} the dwell time half-life. Results of the one-phase exponential fitting are non-significantly different between conditions (ATP_{out} and ATP_{out} + B12_{in}) for both associated and dissociated dwell times. Within the same condition, fitting for associated and dissociated dwell times are significantly different (P < 0.0001 in both ATP_{out} and ATP_{out} + B12_{in}, as calculated using a likelihood ratio test, one-tailed).

3. My final comment in the revision process was to work on the structure of the discussion. I see the authors have worked on that and the structure improved a lot. However, I think that a more thorough refinement of the discussion should still be considered. Subjectively, I found the different parts in the discussion a bit fragmented and I had hard time trying to joint them together in a global picture.

> We acknowledge that we find it difficult to satisfy all readers, but are happy to give it another try if the editor requests so.

Reviewer #3 (Remarks to the Author):

> We thank the reviewer, and hope that it is a useful experience.

Reviewer #4 (Remarks to the Author):

Dear authors,

First, I would like to reiterate that this is a potentially interesting and important study, but still requires a few clarifications to fully support the conclusions.

Several studies (including ones performed by yourselves) had previously shown that for some ECF transporters the S-component and the ECF module dynamically associate/dissociate.

The potential novelty of this study would have been to show that these dynamics are an integral and essential part of the transport cycle, and to the best of my judgement, this has not been demonstrated.

In your response, you explain that performing simultaneous single molecule measurements of association/dissociation and of transport is a daunting task. I fully understand and accept this explanation.

However, there are other, simpler ways, to link the association/dissociation to function. One idea that comes to mind is to X-link the two proteins with linkers of various lengths and flexibility and measure at which point (if ever?!) transport is lost. If your hypothesis is correct, a short and rigid linker would impair transport, while a long and flexible one would not (as has been observed with LacY). This can readily be backed up by sm-FRET measurements, estimating the distance of separation that is required for transport. Maybe no separation, and hence no dissociation is required? For example, there are convincing studies that show a cyclic association-dissociation between the methionine transporter and its SBP (S-component

equivalent). However, other equally convincing studies show that although these dynamics occur, they are perhaps not required for transport.

An alternative to the X-linking approach would be to create a chimera, fused at the genetic level. Similar experiments have been performed with other transport proteins to address such questions.

I FEEL THIS IS A CRITICAL POINT THAT REQUIRES RESOLVING

> We are happy that the reviewer recognizes the value of our work, but we do not agree with the notion that there is a lack of novelty. As we discussed more elaborately above, previous experiments showing that ECF transporter complexes can dissociate have been heavily criticized, because they were done largely with detergent-solubilized transporters, which were considered artifactual. This is why we embarked on our smFRET study, which is novel. There could have been two outcomes of: either the previously criticized detergent-based experiments were artifacts indeed, or they withstand the scrutiny of experimental testing using the more physiologically relevant liposome system. We find the latter, which is hugely important, also as standard for the field – even if it may seem that it had all been done already.

The requested experiments are in our view both beyond the scope of this manuscript, and not the right ones. We emphasize that we *do not claim* in the manuscript that the movement of the substrate molecule across the membrane is strictly dependent on the association-dissociation dynamics of the complex. Although our data strongly suggests such a relationship, we do not claim it exists. For instance, in the discussion we write: “While the results show that dissociation of the S-component takes place in an ATP-dependent way, it does not formally show that expulsion of the S-component is essential for transport to occur.... However, although not formally conclusive, the kinetics of dissociation and association observed here, and the turnover numbers for transport of vitamin B12 determined from ensemble experiments (~0.01 per second) are consistent with the notion that transport and dissociation-association dynamics are linked.”

Obviously, a next step will be to investigate the potential mechanistic coupling (as we state in the manuscript), but this is way beyond the scope of the current manuscript. The suggested experiments would take at least another year to conduct (on top of the four years of work in the current manuscript) and -based on similar experiments that we and others have done in the past- are at best going to be inconclusive, as has been shown repeatedly in the past based on similar fusion and crosslinking experiments. We therefore find the suggested experiments not only out of scope of the current manuscript, but also inappropriate.

This concern is linked to other MAJOR CONCERNS raised by the data:

In the apo state, the complex is stable, with rare dissociation events. Under hydrolyzing conditions, some (low) percentage of the traces become dynamic, with

insignificant difference between ATP (in) and ATP (both sides). To me this could mean only one of two things:

- 1- A significant fraction of the proteins is not active
- 2- The dynamics occur on a timescale faster than the temporal resolution of the measurements.

The confocal experiments argue against option 2, even though, if I understand correctly, these were performed in detergent solution, and the authors go in length explaining why such experiments fall short as they lack the membrane environment and directionality.

> We believe that the reviewer concerns about the percentage of dynamic traces is already addressed in the manuscript. In lines 180-191 we mention the possibility of active transporters not being recorded because the dynamics are slow compared to the length of the movies; the possibility of a fraction of inactive transporters; and the orientation of the transporters in the ATP_{out} condition. For the condition with ATP on both sides, we propose an explanation for the surprising finding that less traces show dynamic FRET compared to the ATP_{out} condition (Lines 198-208). The reviewer is right to point out that we could have missed dynamics faster than the 200 ms time scale used in the TIRF recording, although the absence of dynamics in the recordings in solution and the 0.5s⁻¹ transport rate are against this possibility. We added a sentence in the manuscript line 185.

An additional related concern pertains to the lack of effect of AMP-PNP:

From a thermodynamic viewpoint, if the complex is stable in the apo form, and is “less” stable under hydrolyzing conditions, either the pre-hydrolysis or post-hydrolysis nucleotide bound state needs to be the less stable, dissociated state. In this respect, the data presents two troubling issues:

- 1- To the best of my knowledge, in “regular” bacterial ABC transporters that function as importers, ATP binding, as mimicked here by addition of AMP-PNP, is sufficient for formation of the closed NBDs dimer. Why would then AMP-PNP have no effect on association? From studies of other studies of ABC transporters (importers), the APO state represents one energy level, and the nucleotide bound one state another. Here, these two states seem functionally equivalent. This point is largely ignored by the authors.

> AMP-PNP is a *non-natural* analogue of ATP that indeed in some ABC transporters allows formation of the closed NBD dimer, but this does not happen in ECF transporters, they behave differently. This has been showed in previous work where AMP-PNP bound ECF transporters adopt conformations similar to the apo state with the AMP-PNP molecule mainly interacting with one ATPase binding site (Swier et al 2016, Thangaratnarajah et al 2023 and 2021; references 14, 15 and 16 of the manuscript respectively). Given the structural information, it is not so surprising that ADP and AMP-PNP do not trigger dynamic FRET in our experiments. It should be noted that also in other types of ATPase, AMP-PNP does not always act as ATP.

- 2- If the authors think that the high-energy dissociated state is populated in the post hydrolysis state, this needs to be shown, and can readily be done so by trapping it.

> We do not understand what the reviewer suggests. To arrive at the dissociated state from the associated state the transporters need to go through a high-energy barrier. Trapping a proxy of the high-energy state was not readily done, but rather a big challenge resulting in the publication of the structure of the ECF module bearing the EQ mutations (substituting the catalytic glutamates in the walker B motif) in complex with ATP (pdb structure 8bms, Thangaratnarajah et al 2023). This work was carried out with the ECF module in absence of S-component. The structure corresponds to the schematic of the ECF module in the 'S-component expulsion' step from the Supplementary Figure 1a of the manuscript. The high-energy state is not compatible with an associated S-component, so to trap the complex in the high energy state and purify the full four-subunit complex is not possible. In our model, upon ATP binding and dimer closure a high energy state leads to dissociation of the S-component (low FRET). Upon ATP hydrolysis and dimer opening, the complex remains dissociated and populates a low-energy dissociated state (still low FRET). It is likely that the S-component dissociation is faster than the TIRF recordings resolution as the FRET efficiency transitions happen fully in between two frames.

MINOR COMMENTS

I believe the statistical analysis in 2C need to be done with ANOVA, not with an (two-tailed unpaired t-test

> We performed an unpaired t-test on the weighted average because a simple ANOVA would not take into account the difference in sample size of each repeat (number of traces selected per recording session). But the reviewer is right that ANOVA is appropriate to compare the multiple conditions. We thus performed a Welch ANOVA test adding weight to the repeats (number of traces selected) in order to use weighted average and variance values. This test gave a p-value of 0.00085 confirming that statistically significant differences are present between some of the conditions. To explore further which conditions are significantly different, we conducted a Tukey HSD test and we obtained significant differences between Apo and ATP_{out} (p= 0.000378) and Apo and ATP_{out} + B12_{in} (p = 0.000021). The difference between Apo and ATP_{in & out} was not significant (p = 0.063) as opposed to what was found in the t-test. This might be due to the lower number of repeats in the ATP_{out} + B12_{in}. This result does not affect our interpretation as the condition with ATP on both sides was not explored further for the reasons we mention in the manuscript. The statistical test was details was updated in the manuscript.

The subscript legends in 2b are too small to read and are also confusing. I imagine that "ATP" means ATP on both sides and ATP_{out} means ATP only on the inside. This should be made clear in both the figure and legends.

> We made modifications to improve the readability of the legend of Figure 2b and to clarify conditions in multiple figures. ATP_{out} means ATP only present in the outside buffer and not in the liposome lumen; while ATP_{in & out} indicates the presence of ATP on both sides of the membrane of the liposomes.

"These values are close to the apparent KM value for ATP measured in bulk

transport (190 μM , Supplementary Fig. 8b), suggesting that the dissociation of the complex is the ATP dependent step in transport”

This sentence is quite confusing. Do the authors simply wish to say that these data suggest that indeed dissociation is driven by ATP hydrolysis? If this is the case. I'd suggest saying just that.

This sentence was changed L310.

“Unfortunately, the number of observed transitions within single movies was too low for such analysis, because of the slow transport kinetics (Fig. 2a,c)”

This claim is somewhat misleading since Figure 2A,C show association/dissociation dynamics and propensities and not transport.

This sentence was changed L324.